# RHEA v1.0: Enabling fully coupled simulations with hydro-geomechanical heterogeneity

Jose M. Bastias Espejo[1], Andy Wilkins[2], Gabriel C. Rau[1,3], and Philipp Blum[1]

[1]Karlsruhe Institute of Technology, Institute of Applied Geosciences, Karlsruhe, Germany
[2]Commonwealth Scientific and Industrial Research Organisation (CSIRO), Mining Geomechanics Team, Brisbane, Australia
[3]The University of New South Wales, Connected Waters Initiative Research Centre, Sydney, Australia

**Correspondence:** Jose Bastias (jose.bastias@kit.edu)

**Abstract.** Realistic modelling of tightly coupled hydro-geomechanical processes is relevant for the assessment of many hydrological and geotechnical applications. Such processes occur in geologic formations and are influenced by natural heterogeneity. Current numerical libraries offer capabilities and physics couplings that have proven to be valuable in many geotechnical fields like gas storage, rock fracturing and Earth resources extraction. However, implementation and verification of full heterogeneity of subsurface properties using high resolution field data in coupled simulations has not been done before. We develop, verify and document RHEA (Real HEterogeneity App), an open-source, fully coupled, finite-element application capable of including element-resolution hydro-geomechanical properties in coupled simulations. To extend current modelling capabilities of the Multiphysics Object-Oriented Simulation Environment (MOOSE), we added new code that handles spatial distributed data of all hydro-geomechanical properties. We further propose a simple, yet powerful workflow to facilitate the incorporation of such data to MOOSE. We then verify RHEA with analytical solutions in one and two dimensions, and propose a benchmark semi-analytical problem to verify heterogeneous systems with sharp gradients. Finally, we demonstrate RHEA's capabilities with a comprehensive example including realistic properties. With this we demonstrate that RHEA is a verified open-source application able to include complex geology to perform scalable, fully coupled, hydro-geomechanical simulations. Our work is a valuable tool to assess challenging real world hydro-geomechanical systems that may include different levels of complexity like heterogeneous geology and sharp gradients produced by contrasting subsurface properties.

## 1 Introduction

The complexity of processes occurring in a fluid saturated deformable porous medium and their importance to a wide range of subsurface applications presents a major challenge for numerical modelling especially when including realistic heterogeneity. Example applications in geo-engineering that inherently require coupling of hydro-geomechanical processes are the interaction between pressure, flow and fracturing of rocks (Atkinson, 2015; Weng, 2015; Berre et al., 2019), land surface subsidence caused by the extraction of Earth resources (Peng, 2020; Ye et al., 2016), underground gas storage (Yang et al., 2016; Tarkowski, 2019) and mass movement (Zaruba and Mencl, 2014; Haque et al., 2016; Gariano and Guzzetti, 2016). Even though the fundamental mathematical description of coupled hydro-geomechanical processes has reached general consensus (Cheng, 2016; Wang, 2017), realistic modelling of such processes requires a precise description of the underground.

Heterogeneity is ubiquitous across scales and strongly affects the mechanical properties as well as the movement of fluids through the subsurface. For instance, the hydraulic conductivity of fractures within a porous rock is often orders of magnitude greater than that of unfractured rock, so that fine spatial discretization around fractures is needed in certain numerical models, resulting in expensive computational demands (Morris et al., 2006; Eaton, 2006). As a result, the development of coupled hydro-geomechanical models generally requires simplifying or averaging heterogeneity, i.e. homogenising (Blum et al., 2005,
2009). Recent research has identified the need to improve modelling of coupled hydro-geomechanical systems (Lecampion et al., 2018; Grigoli et al., 2017; Birkholzer et al., 2019), and particularly also the importance of introducing high-resolution details to improve the accuracy of numerical simulations (McMillan et al., 2019). However, integrating spatially distributed material properties to numerical tools is not trivial because the shape of geological formations can consist of complex geometries produced by natural processes acting over a long period.

Terzaghi (1923) first described the elastic interactions between a porous medium and a fluid occupying its pore space, and the unidirectional system's dynamic responses to external forces. Biot (1941) later generalised this theory to three dimensions giving rise to the well-known theory of consolidation or poroelasticity, also termed *Biot* theory. Since the 1970's, a large number of numerical libraries have been developed, optimised and applied to a diverse range of poroelastic applications (Bear and Verruijt, 1987; Verruijt, 1995; Cundall and Hart, 1993; Boone and Ingraffea, 1990). Notable is the work of Verruijt (2013),
who designed a number of numerical solvers for typical one and two dimensional poroelastic problems.

    Well-known subsurface simulation libraries are concisely reviewed in the following. Since the number of subsurface simulation codes is vast, we only included platforms that are relevant to modelling spatially distributed heterogeneity. For an exhaustive list of codes the reader is referred to White et al. (2018). Current subsurface hydro-geomechanical simulation codes can be classified based on the numerical solution scheme and modelling approach of the coupled physics. For example, se-
quential coupling solves for the hydraulic and geomechanical variables independently and in sequence. Notable examples are geomechanics models based on TOUGH (Transport Of Unsaturated Groundwater and Heat) (Pruess et al., 1999; Xu et al., 2006; Lei et al., 2015; Lee et al., 2019). These consist of different libraries to solve for coupled thermo-hydro-mechanical (THM) applications relying on the numerical capabilities provided by TOUGH. The libraries differ in their fundamental equations, numerical solution methods and discretization schemes (Rutqvist, 2017). Although sequential codes allow flexible and
efficient code management in conjunction with reasonable computational costs, they tend to perform poorly in tightly coupled processes, since transient interaction between variables may not be computed accurately (Kim et al., 2011; Beck et al., 2020). However, sequential coupling combined with iterative schemes can significantly improve the numerical accuracy. In such implementations, feedback between variables occurs by transferring hydraulic variables to the geomechanics implementation, followed by returning the calculated stress and strain back into the flow problem for the next iteration (Beck et al., 2020). The
numerical stability of such iterative methods is discussed by Kim et al. (2011); Mikelić and Wheeler (2013). Another massively parallel subsurface flow package is PFLORTRAN, an open-source, multi-scale and multi-physics code for subsurface and surface processes (Hammond et al., 2014). PFLORTRAN solves non-isothermal multi-phase flow, reactive transport and geomechanics in a porous medium. It has previously been applied to simulate hydro-geomechanical systems (Lichtner and Karra, 2014).

Another concept is to solve the hydro-geomechanical equations as a fully-coupled system (i.e. all equations are solved simultaneously). This is often performed using an implicit time-stepping scheme, which has unconditional numerical stability and high accuracy, but is computationally expensive. This approach has proven to be useful in geo-engineering applications (Nghiem et al., 2004; Hein et al., 2016; Pandey et al., 2018). Various fully coupled hydro-geomechanical libraries have been developed and released. Proprietary software such as COMSOL (Holzbecher, 2013) has been used intensively in geomechanical

applications, in particular for modelling of coastal aquifers (Zhao et al., 2017). COMSOL can import material data from text files into simulations. However, an automatic interpolation between neighboring materials is performed automatically and may lead to undesired results. More recently, Pham et al. (2019) included geomechanical and poroelastic capabilities into the proprietary groundwater modelling environment FEFLOW (Finite Element Flow). MRST is an open-source code developed within the proprietary software MATLAB for fast prototyping of new tools in reservoir modelling (Lie, 2019). While MRST

is not a simulator itself, it supports multi-phase flow with THM physics. MRST has been usedfor hydro-geomechanical problems, such as fracture rocks (Zhao and Jha, 2019) as well as several other subsurface applications (Garipov et al., 2018; Ahmed et al., 2017; Edwards et al., 2017). Notably also, two open source Python codes have been developed. The first is the FEniCS project (Haagenson et al., 2020; Alnæs et al., 2015), while the second is called Porepy and was specifically developed to simulate THM processes in rock fractures (Keilegavlen et al., 2017, 2021). Despite the fact that python-based coding offers

the advantage of high-level programming within a relatively friendly user interface, these codes are designed to facilitate rapid development of features that cannot be properly represented by standard simulation tools rather than general multiphysics problems. Other fast-prototyping novel codes include (Dang and Do, 2021; Tran and Jha, 2020; Reichenberger, 2003; Martin et al., 2005; Frih et al., 2012).

An additional option is OpenGeoSys (OGS), a well-known open source library to solve multi-phase and fully coupled

THM physics (Kolditz et al., 2012). The code is well documented and features several examples in different subsurface areas. Further, different developers are constantly contributing new features to the source code (Graupner et al., 2011; Kosakowski and Watanabe, 2014; Li et al., 2014). Similarly, DuMux is a free and open source, fully coupled numerical simulator for multi-phase flow and transport in a porous medium, (Flemisch et al., 2011). It is based on the Distributed Unified Numeric Environments (DUNE), a C++ based ecosystem that solves finite element models based on PETSc. DuMux is well known for its strong focus

on multi-phase flow and transport in a porous medium. Its recent release adds extra features which facilitates physics coupling, such as Navier-stokes models (Koch et al., 2021). From our experience, however, users without some background in computer science, experience in programming in C++ or python as well as using a debugger may require a significant amount of time to take full advantage of the features that OGS and DuMux offer. Furthermore, to our best knowledge we are unaware of a peer-reviewed verification of these codes that includes fully distributed hydraulic and geomechanical heterogeneity.

The multi-physics coupling framework MOOSE (Multiphysics Object Oriented Simulation Environment) (Permann et al., 2020) offers a unique environment where users can couple different physical processes in a modular approach. Within the object-oriented ecosystem of MOOSE, each physical process (or its partial differential equation, PDE) is treated separately as an individual MOOSE object and coupling is performed by the back end routines of MOOSE. The MOOSE numerical scheme is based on the finite element (FE) method. It offers clean and effective numerical PDEs solvers as well as mesh capabilities

with a uniform approach for each class of problem. This design enables easy comparison and use of different algorithms (for example, to experiment with different Krylov subspace methods, preconditioners, or truncated Newton methods) which are under constant development. MOOSE enables the user to focus on describing the governing equations while the underlying numerical technicalities are taken care of by the system.

We have found that mastering the basic concepts of the MOOSE workflow requires a steep learning curve. However, it requires minimum C++ coding skills which facilitates the learning experience from users that not necessarily have a computer science background. Once the basics are mastered the benefits are significant, for example an experienced user can easily modify the source code to add desired features such as multi-scale physics, non-linear material properties, complex boundary conditions or even basic post-processing tools with only a few lines of code.

An example of MOOSE's capabilities in simulating coupled processes in a porous medium was illustrated by Cacace and Jacquey (2017), who developed a MOOSE-based application named GOLEM. It was optimised to model three-dimensional THM processes in fractured rock (Freymark et al., 2019). Another cutting-edge implementation is PorousFlow, an embedded MOOSE library to simulate multi-phase flow and Thermal-Hydraulic-Mechanical-Chemical (THMC) processes in a porous medium (Wilkins et al., 2020). Porous Flow has been verified and applied to simulate a number of complex and realistic systems, for example shallow geothermal systems (Birdsell and Saar, 2020), $CO_2$ sequestration (Green et al., 2018) and groundwater modelling with plastic deformation (Herron et al., 2018). However, it has not yet been extended and verified for the simulation of spatial heterogeneity of mechanical parameters. In other words, despite its ability to handle spatially distributed heterogeneity of permeability and porosity, it does not support spatially distributed heterogeneity of mechanical properties such as bulk and Young's moduli. To the best of the authors' knowledge no existing open source numerical tool is able to integrate full heterogeneity including all hydro-geomechanical parameters representative of complex geologic formations.

The aim of this paper is therefore to develop, verify and illustrate a novel and generic workflow for modelling fully coupled hydro-geomechanical problems allowing the inclusion of hydraulic and geomechanical heterogeneity inherent to realistic geological systems. This was achieved by extending the current capabilities of the MOOSE of two its native physical modules, namely Porous Flow and Tensor Mechanics. We call this workflow RHEA (Real HEterogeneity Application). The name Rhea depicts a flightless bird that is native to the South American continent. RHEA is based on MOOSE's modular ecosystem and combines the capabilities of Porous Flow and Tensor Mechanics with material objects that are newly developed in our work and provide the novel ability to allocate spatially distributed properties at element-resolution in the mesh. By integrating new C++ objects, we modified the underlying MOOSE code within PorousFlow and Tensor Mechanics. To streamline pre-processing efforts arising from this improvement, we developed a Python-based, automated workflow which uses standard data format to generate input files that are compatible with the material objects in MOOSE format. Finally, we verified the correctness of RHEA with a newly developed, analytical benchmark problems allowing vertical heterogeneity and illustrated its performance using a sophisticated 2D example with distributed hydraulic and mechanical heterogeneity. In this work, we first describe the workflow required to compile a RHEA app, formulate a modelling problem and run a simulation. We then compare RHEA's simulation results (verify) with one and two dimensional analytical solutions, and propose a benchmark semi-analytical solution to validate RHEA's performance when sharp gradients are present. Finally, we apply RHEA to a complicated two

dimensional problem with centimetre-scale heterogeneities demonstrating its capabilities. We anticipate that our work will lay the foundation for accurate numerical modelling of hydro-geomechanical problems allowing full spatial heterogeneity.

## 2 Governing equations

Modelling of coupled hydro-geomechanical processes requires solving the equations describing fluid flow in a deformable porous medium. The coupled processes can be described physically in a representative elementary volume (REV) by a balance

of fluid, mass and momentum, where local equilibrium of thermodynamics is assumed and macroscopic balance equations are considered to be the governing equations. In this section, the governing equations for hydro-geomechanical processes in a fully saturated porous medium with liquid fluid are presented on the basis of Biot's theory of consolidation. In the pore pressure formulation, the field variables are the liquid phase pressure $p_f$ and the displacement vector $\boldsymbol{u}$. The material parameters can be spatially variable, but remain independent of time. Permeability and elastic parameters are described as tensors, whereas the

Biot coefficient is a scalar.

Fluid flow within a deformable and fully saturated porous medium is described by the continuity equation

$$\frac{1}{M}\frac{\partial p_f}{\partial t} + \alpha\frac{\partial \varepsilon_{kk}}{\partial t} + \nabla \cdot \boldsymbol{q}_d = Q_f \,, \tag{1}$$

where $\alpha$ is the Biot coefficient, $\varepsilon_{kk}$ the volumetric strain, $Q_f$ a fluid sink or source term and $M$ is the Biot modulus of the porous medium (the reciprocal of the storage coefficient). In Biot's consolidation theory, the Biot modulus is defined as

$$\frac{1}{M} = \frac{\phi}{K_f} + \frac{(\alpha - \phi)}{K_s} \,, \tag{2}$$

where $\phi, K_f, K_s$ represent the porosity, fluid and solid bulk modulus respectively. As Darcy flow is assumed, the fluid discharge $\boldsymbol{q}_d$ can be expressed as a momentum balance of the fluid like

$$\boldsymbol{q}_d = \phi(\boldsymbol{v}_f - \boldsymbol{v}_s) = -\frac{\boldsymbol{k}}{\rho_f \boldsymbol{g}}(\nabla p_f - \rho_f \boldsymbol{g}) \,, \tag{3}$$

where $\boldsymbol{v}_f$ and $\boldsymbol{v}_s$ are the fluid and solid matrix velocities respectively; $\boldsymbol{k}$ is the permeability tensor; $\mu_f$ is the dynamic viscosity

of the fluid; $\rho_f$ is the density of the fluid and $\boldsymbol{g}$ is the gravitational acceleration vector.

The mechanical model is defined via momentum balance in terms of the effective Cauchy stress tensor $\boldsymbol{\sigma'}(x,t)$ as

$$\nabla(\boldsymbol{\sigma'} - \alpha p_f \mathbb{I}) + \rho_b \boldsymbol{g} = 0 \,, \tag{4}$$

where $\mathbb{I}$ is the rank-two identity tensor. The mass of fluid per volume of porous medium is expressed as the sum of the phases

$$\rho_b = \phi\rho_f + (1 - \phi)\rho_s \,, \tag{5}$$

where $\rho_s$ is the solid density. The elastic strain can be expressed in terms of displacements with the relation

$$\varepsilon = \frac{1}{2}(\nabla\boldsymbol{u} + \nabla^T\boldsymbol{u}) \,. \tag{6}$$

The effective stress is related to elastic strains by the generalized Hooke's law:

$$\varepsilon = \varepsilon_{ij} = \mathbb{C}_{ijlk}\sigma'_{ij} \; , \tag{7}$$

where $\mathbb{C}_{ijkl}$ is the elastic compliance tensor.

Together, Eqs. 1 to 7 constitute the coupled system that represents hydro-geomechanical systems with linear elastic deformation.

As a derivative of the MOOSE framework, RHEA enables access to a wide array of options to fine tune a simulation. Solver options such as numerical schemes, adaptive time-stepping as well as general PETSc options are available. By default, RHEA uses a first order fully-implicit time integration (backward Euler) for unconditional stability and solves the coupled equations simultaneously (full coupling) (Kavetski et al., 2002; Manzini and Ferraris, 2004; Gaston et al., 2009). RHEA also allows operator splitting to implement loose coupling, i.e., solving the fluid flow while keeping the mechanics fixed, then solving the mechanics while keeping the fluid-flow fixed. While this can be executed on separate meshes with different time-stepping schemes, this feature is not explored in the current article (Martineau et al., 2020).

Explicit time integration (with full or loose coupling) and other schemes such as Runge-Kutta are available in MOOSE and RHEA, but stability limits the time-step size, so these are rarely used in the type of subsurface problems handled by RHEA. By default, MOOSE and RHEA use linear Lagrange finite elements (tetrahedra, hexahedra and prisms for 3D problems, triangles and quads for 2D problems), but higher-order elements may be easily chosen if desired (Hu, 2017).

RHEA does not implement any numerical stabilization for the fluid equation to eliminate overshoots and undershoots, however, fluid volume is conserved at the element level (Cacace and Jacquey, 2017). Although not explored in this article, RHEA's fluid flow may be extended to multi-phase, multi-component flow with high-precision equations of state, as well as finite-strain elasto-plasticity (Wilkins et al., 2020).

## 3 Building RHEA

Real Heterogeneity App (RHEA) is an open-source simulation workflow and tool specifically developed to allow fully coupled numerical simulations in a saturated porous medium with spatially distributed heterogeneity in hydraulic and geomechanical properties. We built RHEA as a derivative of MOOSE, the massively parallel and open source FE simulation environment for coupled multi-physics processes (Gaston et al., 2009; Permann et al., 2020). MOOSE offers virtually unlimited simulation capabilities covering a wide spectrum of applications. This is based on a workflow where the end user does not need to know the details of the FE implementation. To achieve that, MOOSE utilises the libMesh library, a framework capable of manipulating multi-scale, multi-physics, parallel and mesh-adaptive FE simulations (Kirk et al., 2006). While the numerical methods, solvers and routines are executed by PETSc libraries (Balay et al., 2019), MOOSE is designed to allow the user to interact and control these two libraries without having to do any complex programming. Instead, the user frames the problem simply through an input file with unique syntax.

We found that learning how to perform numerical simulations based on the MOOSE framework is not a trivial task. Our aim is to further develop modelling capabilities while simplifying the complexity of the problem through an easy to follow workflow accompanied by a visual summary. The RHEA workflow can be summarised as follows:

**Step 1 - RHEA compilation:** The user creates the RHEA application following the structure outlined in Fig. 1. In other words, the user creates an executable file which is able to model fully coupled hydro-gemechanical systems in a heterogenous medium. We accomplished this through new MOOSE-based materials functions able to allocate data in each element of the mesh based on a pre-generated input file. Furthermore, we integrated the multi-physics of Section 2 to RHEA by adding the Porous Flow (Wilkins et al., 2020) and Tensor Mechanics modules that are part of the MOOSE framework. Once RHEA is downloaded, the user can access the necessary files to build RHEA, and can even access those files to modify the physics. This procedure is generic for any new MOOSE application. The core components of any MOOSE app such as RHEA are (Fig. 1):

**Block 1 - Kernels:** The *kernels* (or partial differential equation terms) describing the physics are implemented in their weak form (Jacob and Ted, 2007). In the MOOSE ecosystem, PDEs are represented by one simple line of code, this is highlighted with a cyan rectangle in Fig. 1. This straightforward way of describing complex multi-physics constitutes the most powerful feature of MOOSE.

**Block 2 - Material properties:** Values, including spatially-distributed values can be prescribed for each of the materials appearing in the *kernels*.

**Block 3 - Kernel coupling:** The user can couple different physics by including different *kernels* in its model, or by creating new *kernels*.

This dynamic procedure allows flexible creation of the RHEA application or any MOOSE-based application requiring minimal knowledge of C++ programming skills.

**Step 2 - Preparation of material properties:** The spatially distributed data is formatted to the structure required by the RHEA app compiled in Step 1. We implemented this with a custom Python script that imports and formats the original CSV or VTK dataset into a RHEA-compatible data structure. Within RHEA, the hydro-geomechanical material properties are field properties which means that each value in the data set has to be allocated to a respective mesh element. Therefore, when the mesh is generated, the discretization has to match the number of data points of the data set. That way, each property value is represented within the simulation. Note that if this is not done correctly, RHEA may assign undesired property values. This is because RHEA will automatically linearly interpolate any values provided to the mesh. Thus, if the initial mesh discretization does not match the user-supplied samples, interpolated values are assigned which could lead to undesired results.

**Step 3 - Simulation setup:** To define the numerical model, a RHEA script has to be created in the standard MOOSE syntax. The script consists of an array of systems that describe the mesh, physics, boundary conditions, numerical methods and

outputs. A short example along with brief system descriptions is illustrated in Fig. 1. The blocks consist of MOOSE
functions that are written and design in a generic manner and independently of the nature of the problem, this way the
blocks can be recycled and reused. The spatially distributed material properties can be imported into the *Function* system
and subsequently be stored in the *AuxVariable* system to be assigned as material property in the *Materials* block.

     In summary, numerical simulations of hydro-geomechanical problems with spatially distributed material properties can be
performed by calling RHEA's executable file (created in Step 1) using the simulation control script (created in Step 3) which
contains the necessary instructions as well as reading in the spatially distributed material properties (created in Step 2).

## 4   Verifying RHEA

To test if RHEA accurately solves the differential equations stated in Sect. 2 and if boundary conditions are correctly satisfied,
four different tests were developed. The proposed tests use predefined material properties that were imported into RHEA using
the workflow presented in Sect. 3. The tests were designed to gradually build up complexity and cover the typical spectrum
of consolidation problems. In two of the examples, RHEA's performance in simulations with sharp gradients is tested. First,
a one dimensional consolidation problem where the hydraulic conductivity varies in four orders of magnitude between layers
and a second two dimensional example with realistic heterogeneity in which the hydraulic conductivity of the geological facies
varies over six orders of magnitude.

The first test, the classical Terzaghi's problem, is used as a basic benchmark of the hydro-mechanical coupling in RHEA.
In later sections, we illustrate the full potential of RHEA when simulating spatially-heterogeneous systems in one and two
dimensions. The four verification scenarios are described in the following subsections. All numerical solutions were calculated
using an 8 core Intel i7-3770 CPU @ 3.40GHz with 32 GB DDR4 RAM memory and the results were stored on a hard disk
drive.

**4.1   Terzaghi's problem**

In the one dimensional consolidation problem, also known as Terzaghi's problem (Terzaghi, 1923), a single load $q$ is applied
at $t = 0$ on the top of a fully saturated homogeneous sample with the height $L$. The system is only drained at the top, where the
pressure of the fluid is assumed to be $p = 0$ for $t > 0$. At the moment of loading, $t = 0$, the undrained compressibility of the
solid increases the pressure of the sample. For $t > 0$, the system is allowed to drain and the consolidation processes begins.
In the absence of sources and sinks, Eq. 1 is reduced to the basic storage equation as

$$\frac{1}{M}\frac{\partial p_f}{\partial t} + \alpha\frac{\partial \varepsilon_{zz}}{\partial t} = \frac{k}{\gamma_f}\frac{\partial^2 p_f}{\partial z^2}, \tag{8}$$

where the product $\rho_f \cdot g$ was written as $\gamma_f$ and represents the volumetric weight of the fluid. Eq. 3 is used to couple the fluid
discharge $\boldsymbol{q}_d$. From Hook's law, assuming one-dimensional deformation, the vertical strain equals the volume change

$$\frac{\partial \varepsilon_{zz}}{\partial t} = -m_v\frac{\partial \sigma'_{zz}}{\partial t} = -m_v\left(\frac{\partial \sigma_{zz}}{\partial t} - \alpha\frac{\partial p_f}{\partial t}\right), \tag{9}$$

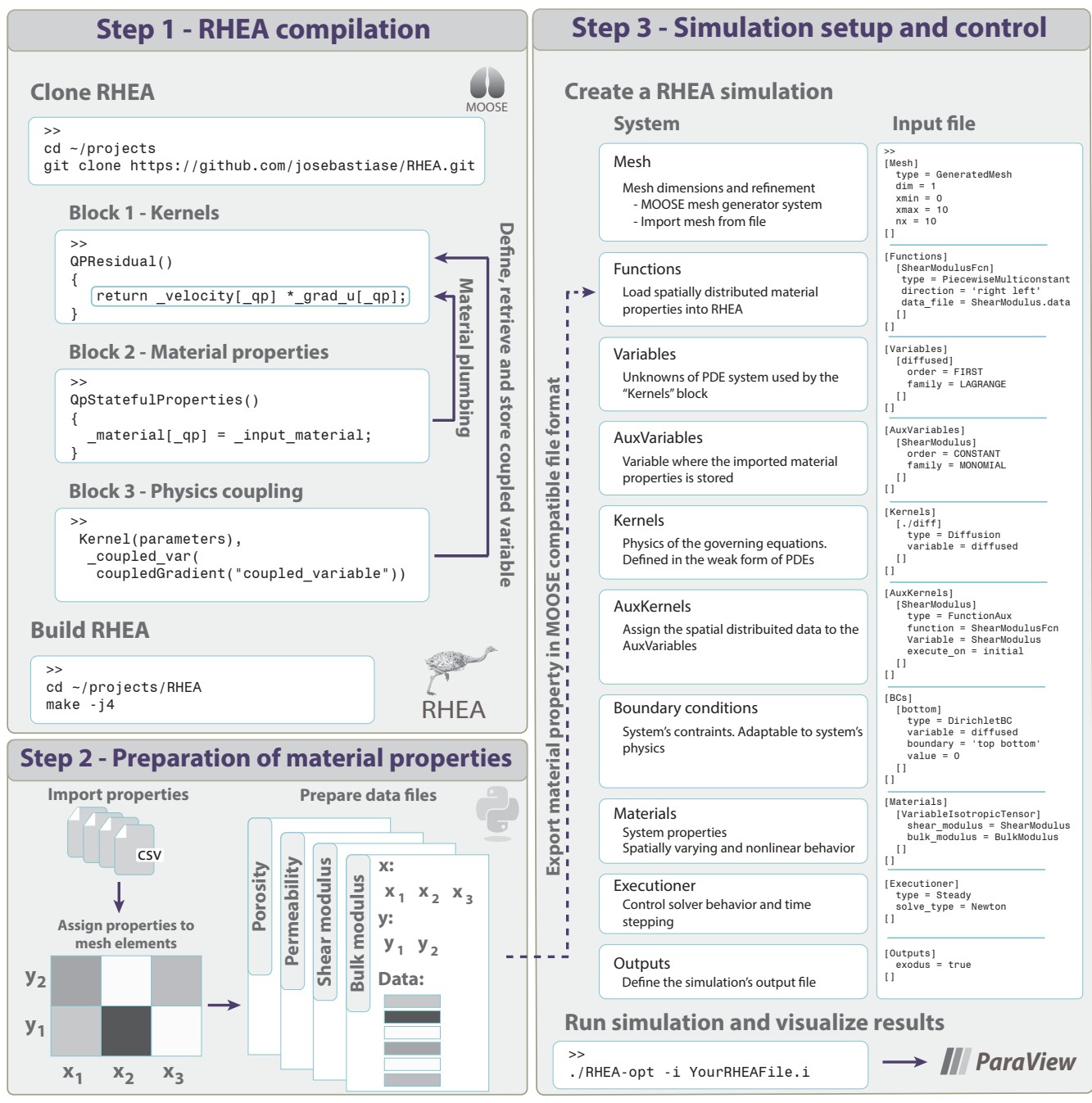

**Figure 1.** Visual illustration of the steps required to create RHEA, generate distributed material properties files and write a simulation script.

where $m_v$ is the confined compressibility of the porous medium

$$m_v = \frac{1}{K_s + (4/3)G_s} \tag{10}$$

and $K_s$ and $G_s$ are the bulk and shear moduli of the porous medium respectively. Substituting Eq. 9 into the storage equation (Eq. 8), the general differential equation for one dimensional consolidation is obtained:

$$\frac{\partial p_f}{\partial t} = \frac{\alpha m_v}{(1/M + \alpha^2 m_v)} \frac{\partial \sigma_{zz}}{\partial t} + \frac{k}{\gamma_f (1/M + \alpha^2 m_v)} \frac{\partial^2 p_f}{\partial z^2} \tag{11}$$

For $t > 0$, the total load $q$ is kept constant and the total stress $\sigma_{zz}$ is also constant. Consequently, Eq. 11 reduces to

$$t > 0 : \frac{\partial p_f}{\partial t} = \frac{k}{\gamma_f (1/M + \alpha^2 m_v)} \frac{\partial^2 p_f}{\partial z^2}. \tag{12}$$

Since the system is undrained at $t = 0$, the initial condition can be established from Eq. 11 as

$$t = 0 : p_f = p_0 = \frac{\alpha m_v}{(1/M + \alpha^2 m_v)} q. \tag{13}$$

The boundary conditions at the top and bottom of the sample are

$$t > 0, z = L : p_f = 0 \tag{14}$$

and

$$t > 0, z = 0 : \frac{\partial p_f}{\partial z} = 0. \tag{15}$$

The analytical solution of the problem is well known and reads (Wang, 2017; Cheng, 2016; Verruijt, 2018)

$$\frac{p_f}{p_0} = \frac{4}{\pi} \sum_{k=1}^{\infty} \frac{(-1)^{k-1}}{2k-1} \cos\left[(2k-1)\frac{\pi}{2}\frac{z}{L}\right] \exp\left[-(2k-1)^2 \frac{\pi^2}{4} \frac{kt}{\gamma_f (1/M + \alpha^2 m_v) L^2}\right]. \tag{16}$$

For this example, the height of the sample was set to $100\,\mathrm{m}$, the hydraulic conductivity is $1 \cdot 10^{-4}\,\mathrm{m/s}$, the porosity is $0.2$, the Biot coefficient is $0.9$, the bulk modulus is $8.40 \cdot 10^7\,\mathrm{Pa}$ and the shear modulus is $6.25 \cdot 10^7\,\mathrm{Pa}$. The performance and consistency of RHEA on the consolidation problem is shown as pore pressure versus depth profiles at discrete times in Fig. 2a. A comparison of the analytical and RHEA's solution reveals excellent agreement, thereby verifying the numerical solution. The total time for computing 101 time steps was 1.92 s.

## 4.2 Layered Terzaghi's problem

The objective of this test is to investigate the performance of RHEA when heterogeneity and sharp gradients are present. The consolidation experiment of the previous section is performed on a sample with multiple layers of contrasting properties. For simplicity, porosity and mechanical parameters are assumed homogeneous. Since the total load $q$ is constant for $t > 0$, Eq. 11 reduces to Eq. 12 across $n$ layers as follows

$$t > 0 : \frac{\partial p_{fi}}{\partial t} = \frac{k_i}{\gamma_f (1/M + \alpha^2 m_v)} \frac{\partial^2 p_{fi}}{\partial z^2}, \quad i \in [1, n], \tag{17}$$

which describes the consolidation in each layer. Here, $z_{i-1} \le z \le z_i$ is the depth of the sample, $p_{fi}$ and $k_i$ are the fluid pressure and permeability of the solid in each layer $i$, respectively. The contact between layers is assumed to be perfect, i.e. the boundary conditions at the layers is represented by equivalent matching fluid pressure as

$$t > 0, z = z_i : k_i \frac{\partial p_{fi}}{\partial z} = k_{i+1} \frac{\partial p_{fi+1}}{\partial z}. \tag{18}$$

The sample is drained at the top, whereas the bottom remains undrained

$$t > 0, z = z_0 = L : p_f = 0 \tag{19}$$

and

$$t > 0, z = z_n = 0 : \frac{\partial p_f}{\partial z} = 0. \tag{20}$$

The fluid pressure produced by the external load starts to dissipate when $t > 0$, but at different rates depending on the consolidation coefficient of the layer. The height of the sample is $100\,\mathrm{m}$ and 10 layers are equally distributed along the sample with $10\,\mathrm{m}$ height. To represent sharp gradients, the selected hydraulic conductivities have four orders of magnitude difference between layers, $1 \cdot 10^{-4}\,\mathrm{m/s}$ and $1 \cdot 10^{-8}\,\mathrm{m/s}$. The high and low permeability layers are alternating. The porosity is set to $0.2$, the Biot coefficient is $0.9$, the bulk modulus is $8.40 \cdot 10^7\,\mathrm{Pa}$ and the shear modulus is $6.25 \cdot 10^7\,\mathrm{Pa}$.

A step-by-step semi-analytical solution of the diffusion problem in a layered sample was derived by Hickson et al. (2009). To solve this problem in RHEA, a mesh of 100 elements was used with a time step of $1 \cdot 10^4$ s. The total time for computing 701 time steps was 13.8 s. A comparison between the analytical solution and RHEA's numerical simulation is shown in Fig. 2b. In the layers with high hydraulic conductivity, the consolidation process occurs rapidly leading to faster pore pressure dissipation (vertical pore pressure profile), and therefore also faster water movement. In contrast, the consolidation process is slower in the low conductivity layers with slower pore pressure dissipation and water movement.

## 4.3 Plane strain consolidation

To evaluate the performance of RHEA for two-dimensional heterogeneity, a consolidation problem with plane strain is developed. The two-dimensional consolidation caused by a uniform load over a circular homogeneous area can be represented by the storage equation (Eq. 1) in two dimensional case as

$$\frac{1}{M} \frac{\partial p_f}{\partial t} + \alpha \frac{\partial \varepsilon}{\partial t} = \frac{k}{\gamma_f} \left( \frac{\partial^2 p_f}{\partial x^2} + \frac{\partial^2 p_f}{\partial z^2} \right) \tag{21}$$

where $\varepsilon$ represents the volumetric strain. Including two equilibrium equations, in terms of total stress, as

$$\frac{\partial \sigma_{xx}}{\partial x} + \frac{\partial \sigma_{zx}}{\partial z} = 0 \tag{22}$$

and

$$\frac{\partial \sigma_{xz}}{\partial x} + \frac{\partial \sigma_{zz}}{\partial z} = 0. \tag{23}$$

The total stress is related to the effective stress through

$$\sigma_{xx} = \sigma'_{xx} + \alpha p \qquad \sigma_{xz} = \sigma'_{xz} \tag{24}$$

and

$$\sigma_{zz} = \sigma'_{zz} + \alpha p \qquad \sigma_{zx} = \sigma'_{zx}. \tag{25}$$

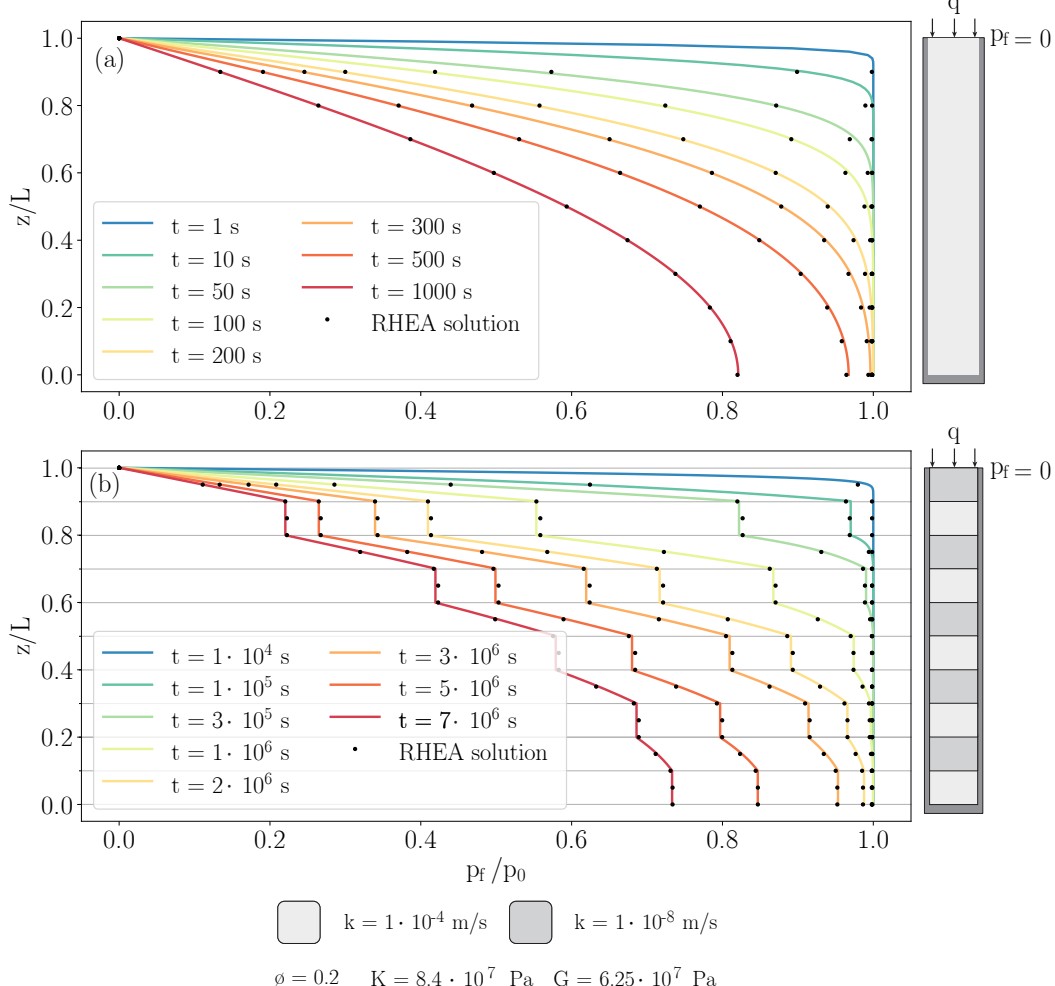

**Figure 2.** The lines represent the analytical solution whereas the dots represent the RHEA solution. (a) Homogeneous case. For this simulation, a total of 100 nodes and 99 elements were set. (b) Heterogeneous case. For this simulation, a total of 100 nodes and 99 elements were set.

The analytical solution can be found by expressing the equilibrium Eq. 22 and Eq. 23 in terms of the displacement components $u_x$ and $u_z$ using Hooke's law as

$$\sigma'_{xx} = -\left(K_s - \frac{2}{3}G_s\right)\varepsilon - 2G_s\frac{\partial u_x}{\partial x} \tag{26}$$

and

$$\sigma'_{zz} = -\left(K_s - \frac{2}{3}G_s\right)\varepsilon - 2G_s\frac{\partial u_z}{\partial z} \tag{27}$$

and

$$\sigma'_{xz} = \sigma'_{zx} = -G_s \left( \frac{\partial u_z}{\partial x} + \frac{\partial u_x}{\partial z} \right). \tag{28}$$

Here, the assumed plane strain is the $y$ axis, i.e. $u_y = 0$. Replacing Hooke's law in plane strain (Eq. 26 to 28) with the effective stress balance (Eq. 22 and Eq. 23 combined with Eq. 24 and Eq. 25) leads to a complete system of equations as

$$\left( K_s + \frac{1}{3} G_s \right) \frac{\partial \varepsilon}{\partial x} + G_s \nabla^2 u_x - \alpha \frac{\partial p_f}{\partial x} = 0 \tag{29}$$

and

$$\left( K_s + \frac{1}{3} G_s \right) \frac{\partial \varepsilon}{\partial z} + G_s \nabla^2 u_z - \alpha \frac{\partial p_f}{\partial z} = 0, \tag{30}$$

where the elastic strain is

$$\varepsilon = \frac{\partial u_x}{\partial x} + \frac{\partial u_z}{\partial z}. \tag{31}$$

The boundary conditions are represented by a constant load in an area of width $2a$, applied at $t = 0$. The system is allowed to drain for $t > 0$ as

$$t > 0, \, z = 0 : \, p_f = 0 \tag{32}$$

and

$$t > 0, \, z = 0 : \, \sigma_{zz} = \begin{cases} q, & |x| < a \\ 0, & |x| > a \end{cases} \tag{33}$$

and

$$t > 0, \, z = 0 : \, \sigma_{xz} = 0. \tag{34}$$

When the sample is loaded, a confined pore pressure in generated which starts to drain instantaneously through the borders of the system. A semi analytical solution in the Fourier domain and Laplace transform for the given equation system and boundary conditions is presented in Verruijt (2013). The height and the width of the sample are $10 \, \mathrm{m}$. The load is applied on the surface of the sample between $-1$ and $1 \, \mathrm{m}$. The hydraulic conductivity is $1 \cdot 10^{-5} \, \mathrm{m/s}$, the porosity is 0.2, the Biot coefficient is 0.9, the bulk modulus is $8.40 \cdot 10^7 \, \mathrm{Pa}$ and the shear modulus is $6.25 \cdot 10^7 \, \mathrm{Pa}$. To solve this problem, a coarse mesh was defined, and MOOSE's native mesh adaptivity system was employed to automatically generate a finer resolution in areas where the pore pressure gradients are steep. This significantly reduces the computational time when compared with using a fine mesh throughout. The total simulation time was 312.6 s for 101 time steps and 10.000 elements with 20.502 nodes.

The results are illustrated in two figures, Fig. 3a shows a cross section of the sample as contour plot. Figure 3b shows a pore pressure profile with depth at the center of the sample $x = 0 \, \mathrm{m}$. Excellent agreement between the analytical solution and the simulated solution by RHEA is observed.

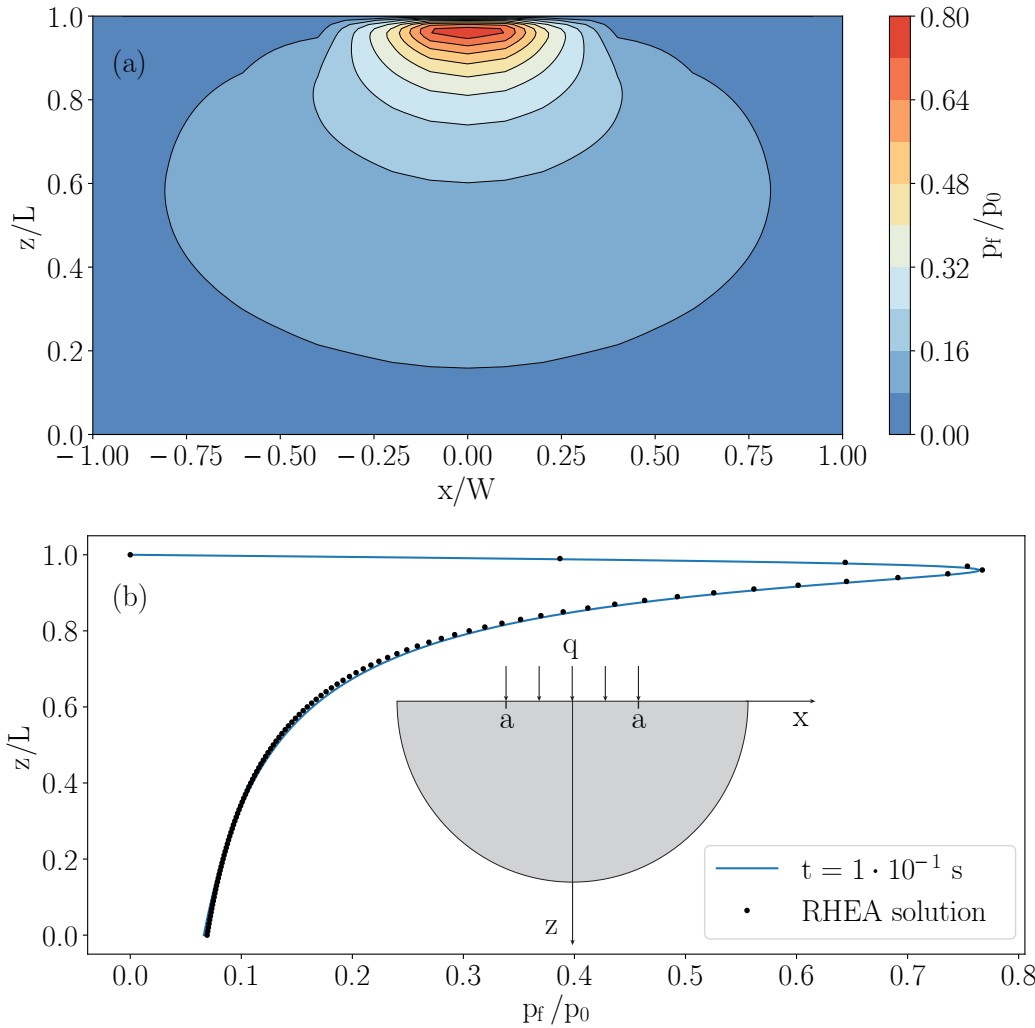

**Figure 3.** The solution of the consolidation problem in plane strain by RHEA is shown in a sample $10\,\mathrm{m}$ wide and $10\,\mathrm{m}$ of height. (a) Contour plot of the solution at time $1 \cdot 10^{-2}$ s. (b) A comparison of the semi-analytical solution (continuous line) and the RHEA solution (dotted line). The differences in pore pressure between both approaches is due to the assumption of an infinite domain in the analytical solution which is not feasible to replicate the latter with RHEA.

## 340  4.4  Modelling realistic geology

The last example aims to study and illustrate the performance of RHEA's with a real data set. This example illustrates how to generate input files using the developed workflow and demonstrates the potential of RHEA for simulating increased spatial complexity and sharp gradients. While the Herten analog is a 3D data set, the example was reduced to two dimensions to facilitate visualization. However, simulations in three dimensions are also possible and can be done using the presented

workflow in unmodified form. The 2D consolidation problem was solved with RHEA, integrating the multi-facies realizations and material properties of the Herten analog (Bayer et al., 2015). Although the data set does not contain geomechanical sub-surface properties, the hydraulic conductivity varies over six orders of magnitude which provides sufficient proof of RHEA's capabilities.

### 4.4.1    Herten aquifer dataset description

Realistic modelling relies not only on accurate data concerning material parameters, but also on appropriate spatial distribution of such parameters (Houben et al., 2018; Irvine et al., 2015; Kalbus et al., 2009). Typically, distributed material parameters are generated by stochastic random fields based on an *a priori* statistical distribution (Vanmarcke et al., 1986). Although random fields have proven to be useful, they do not capture the usual continuity of material parameters (Strebelle, 2002). Consequently, the use of high resolution data, such as "aquifer analogs", is preferred (Alexander, 1993; Zappa et al., 2006).

Aquifer analogs consist of centimeter-resolution data obtained from detailed investigation of geological formations at outcrops. Although aquifer analogs are rare, they have been widely used in different subsurface fields (Höyng et al., 2015; Beaujean et al., 2014; Finkel et al., 2016). The Herten analog is a well-known and rigorously generated 2D outcrop (Bayer et al., 2015). It consists of a fluvial braided river deposit from the south east of Germany, which represents one of the most important drinking water resources in central Europe. Its architecture consists of sedimentary facies, and its body of unconsolidated gravel and

well sorted sand. The dimensions of the 2D outcrop are $16$ m wide by $7$ m high, and features horizontal and vertical data resolution of $5$ cm for hydraulic conductivity and porosity. Hence, the 2D cross-section has a total of $4480$ measurements points. The corresponding hydraulic conductivity $k$, ranges from $6.0 \cdot 10^{-7}$ m/s to $1.3 \cdot 10^{-1}$ m/s, and porosity $\phi$, from $0.17$ to $0.36$ (Fig. 4a). To represent spatial distribution of mechanical properties, typical values of bulk and shear moduli for gravel and sand were assumed to be linearly correlated with the porosity of the aquifer: similar trends have been reported in previous

studies (Mondol et al., 2008; Hardin and Kalinski, 2005; Hicher, 1996). Representative geomechanical moduli can be found in soil mechanics literature as shown in Table 1. The elastic tensor is assumed isotropic in this example, hence elastic moduli are related via (Wang, 2017; Cheng, 2016)

$$
\begin{aligned}
K_s &= \frac{E_s}{3(1 - 2\nu_s)} \\
G_s &= \frac{E_s}{2(1 + \nu_s)} \,,
\end{aligned}
\tag{35}
$$

where $E_s$ and $\nu_s$ denote the Young's modulus and Poisson's ratio of the solid material respectively. The result is that the bulk

moduli vary between $6.7 \cdot 10^7$ Pa and $1.7 \cdot 10^8$ Pa, whereas the shear moduli range between $3.0 \cdot 10^7$ Pa and $3.5 \cdot 10^8$ Pa, as shown in Fig. 4a. RHEA does not require the mechanical moduli to be related to the hydraulic properties in the way we have described in this particular example.

| Material | Young's modulus (MPa) | Poisson's ratio (-) | Reference |
|---|---|---|---|
| Loose gravel | 48 - 148 | - | (Subramanian, 2011) |
| Dense gravel | 96 - 192 | - | (Subramanian, 2011) |
| Gravel | 50 - 100 | 0.3 - 0.35 | (Look, 2014) |
| Sand and gravel | 69.0 - 172.5 | 0.15 - 0.35 | (Das, 2019) |
| Gravel | 68.9 - 413.7 | 0.4 | (Xu, 2016) |
| Dense sand | - | 0.3 - 0.4 | (Lade, 2001) |
| Loose sand | - | 0.1 - 0.3 | (Lade, 2001) |
| Gravel | - | 0.1 - 0.4 | (Kulhawy and Mayne, 1990) |

**Table 1.** Typical elastic properties of sand and gravel.

### 4.4.2 Problem and model description

The two dimensional consolidation is described by Eqs. 21, 29 and 30. A constant load at the top of the sample is applied at $t > 0$, which generates a confined pore pressure. After that, the system is allowed to drain through the top boundary and is subjected to the normal stress. The sample's bottom and sides are impermeable to the fluid, and subject to roller boundary conditions.

For this simulation, a quadrilateral mesh was generated with the mesh generator system of MOOSE. The mesh has $44.800$ elements and $44.940$ nodes, which matches the data set resolution. Since the material properties of the data set differs in orders of magnitude, the mesh adaptivity system of MOOSE was used to ensure accurate results. At each time step the $30\%$ of elements with the highest pore pressure gradient were refined, which reduces the local error at contrasting facies. Hence the mesh is refined in each time step. At the end of the simulation, the number of nodes had grown to $708.548$ and the number of elements to $631.615$. The total simulation time was 0.49 hours for 70 time steps and 44,341 elements with 44,800 nodes.

### 4.4.3 Simulation results

The pore pressure profiles depicted in Fig. 5 illustrate how the physical heterogeneity of the cross-bedded data set strongly influences the fluid flow through the sample. The effect of the centimeter resolution of the data set can be studied when the initial load is applied at $t = 0$. Since the sample is not yet allowed to drain, confined porepressure is generated which depends on the geomechanical characteristics of the sand and gravel. In facies where the soil is highly compressible, the generated pore pressure is also relatively high since the total load is shared between the the fluid and the soil. In contrast, in facies that have higher elastic moduli values the confined pore pressure is relatively low. This effect is nicely shown in Fig. 5a. At time $t > 0$ the top of the sample is allowed to drain. The effect of the highly permeable units made of poorly sorted and well sorted gravel is shown in Fig. 5b. The top facie of the aquifer consists of a highly permeable soil ($k = 1.3 \cdot 10^{-1}$ m/s), which is divided by a thin low permeable layer ($k = 6.1 \cdot 10^{-5}$ m/s), the latter causes contrasting pore pressure profiles. Similar permeability effects

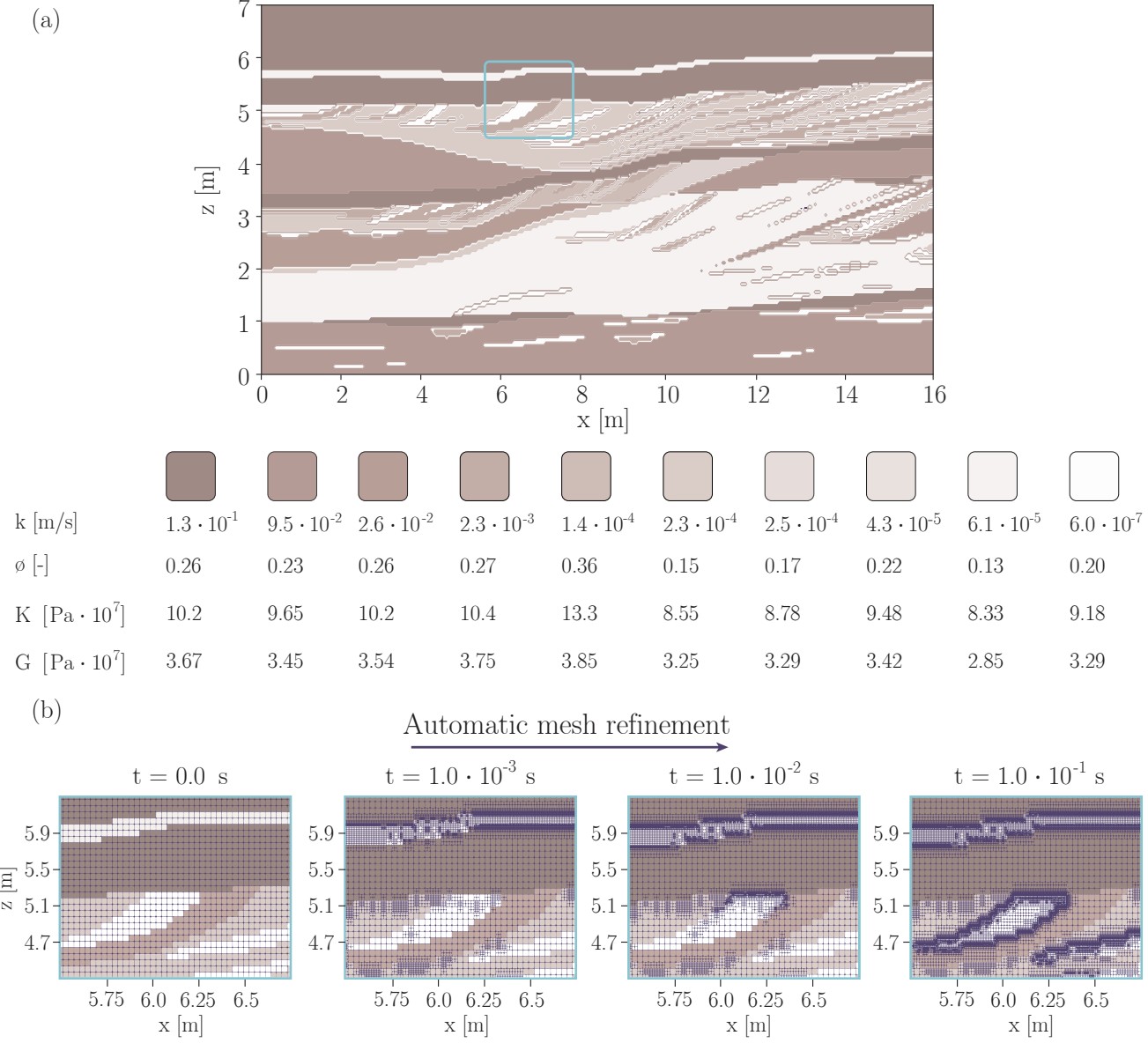

**Figure 4.** Facie architecture and properties of the Herten aquifer analog. (a) Color scale of the hydro-geomechanical properties of the aquifer imported to RHEA. (b) Shows the mesh discretization and its dynamic evolution when the mesh adaptivity system is activated. The time evolution is shown from left to right.

have been discussed before (Choo and Lee, 2018; Peng et al., 2017; Kadeethum et al., 2019). The influence of the temporal and spatial scales on the consolidation process is shown in Fig. 5c and 5d. It can be observed that the process occurs rather

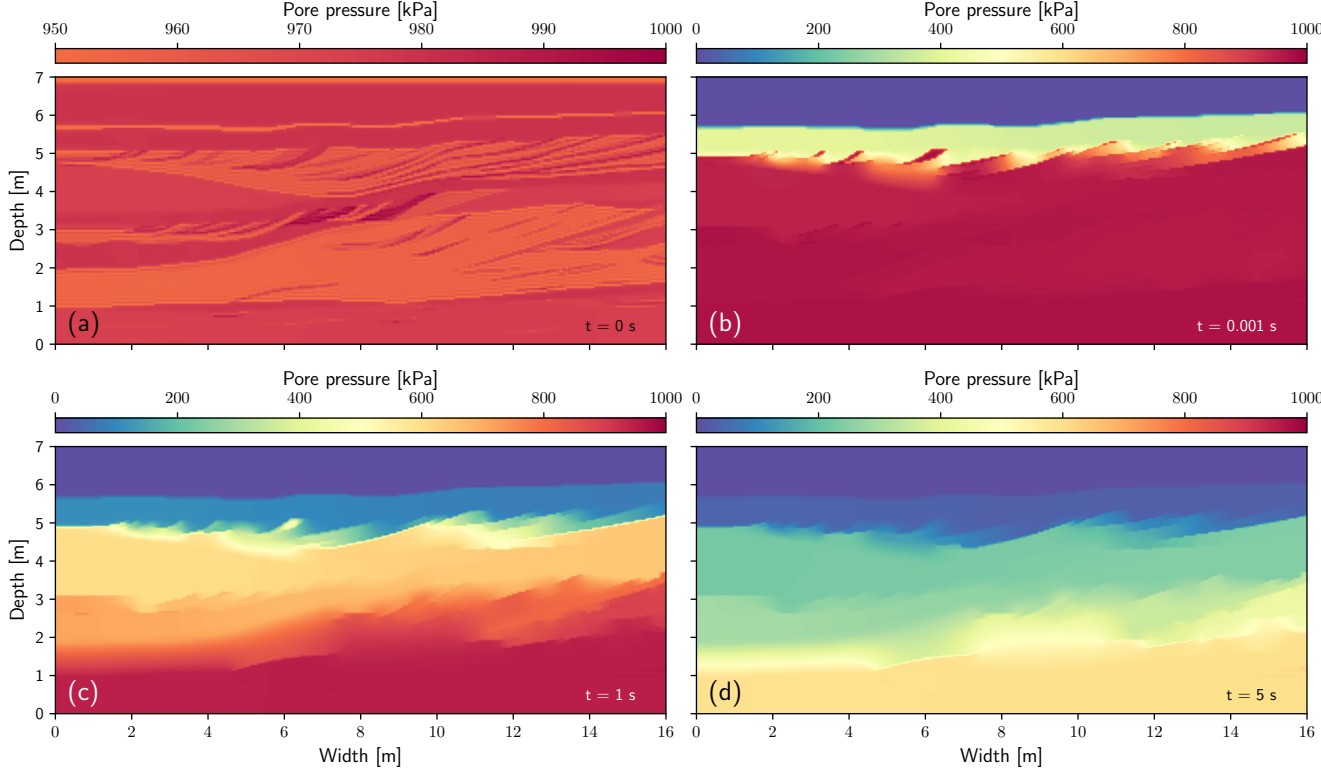

**Figure 5.** Sequence of snapshots of the consolidation process and pore pressure variation in the aquifer with time. (a) Displays the initial condition of the simulation. (b) Snapshot at time $1 \cdot 10^{-3}$ s. (c) Snapshot at time 1 s. (d) Snapshot at time 5 s. Note that (a) uses a different colour range to highlight the small variations in pore pressure.

quickly and is strongly influenced by the low permeability facies. This example demonstrates that RHEA can solve complex and realistic heterogenous hydro-geomechanical coupled problems.

## 5    RHEA's potential

In this paper we develop and verify Real HEterogeneity App (RHEA): a numerical simulation tool that allows fully coupled
numerical modelling of hydro-geomechanical systems. Moreover, RHEA can easily include full heterogeneity of parameters as occurs in real subsurface systems. RHEA is based on the powerful Multiphysics Object-Oriented Simulation Environment (MOOSE) open source framework. Furthermore, we provide an easy-to-understand workflow which explains how to compile the application and run a customised numerical simulation. Despite its simplicity, the workflow combines all the technical advantages provided by MOOSE and its well established framework. The latter allows the development and use of state-of-
the-art and massively scalable applications backed by the unconditional support of a growing community.

Beyond unlocking the ability to include full heterogeneity of hydro-geomechanical parameters in simulations, our contribution provides examples to verify future numerical codes. Additionally, a semi-analytical benchmark problem is proposed to verify the performance of numerical code when heterogeneity and sharp gradients are present.

Our example simulations illustrate that the subsurface hydro-geomechanical properties, in particular permeability (or transmissivity), play a key role in the consolidation process. Although this insight is valuable, it can lead to an oversimplification when models assume transmissivity varies heterogeneously while mechanical parameters are assumed homogeneous. This approach can lead to biased results in systems where different geologic formations are present. For example, land surface subsidence is a process that can occur due to anthropogenically induced decrease of subsurface pore pressure causing progressive consolidation and slow downward percolation across the layers within the subsurface. This process depends on the spatial distribution of the geomechanical properties, in particular those of clay layers within the subsurface. RHEA could be used to increase our understanding of the spatial and temporal evolution of land surface subsidence. Our newly developed workflow enable such advanced numerical simulations.

RHEA has the potential to advance our understanding of real world systems that have previously been oversimplified. Further, RHEA offers the integration of high resolution data set with sophisticated numerical implementations. Potential numerical instabilities caused by highly heterogeneous systems (i.e. settings with sharp gradients) are handled automatically by combining adaptive meshing capabilities with implicit time stepping. While this work demonstrates RHEA's capabilities for two dimensional problems, this can easily be extended to three dimensional simulations. In that case, a three dimensional mesh that is representative of the spatially distributed hydraulic and geomechanical properties of any available dataset can be generated. The tasks follow the data formatting workflow and simulation control as described in Section 3.

Our current work focuses on hydro-geomechanical coupling of heterogeneous systems. However, RHEA could potentially be extended to include also thermal processes and three dimensional simulations. While it would allow fully coupled simulations of thermal-hydraulic-mechanical (THM) systems including spatially distributed heterogeneities, verification will require the development of more advanced analytical solutions, a task that however is beyond the scope of this contribution.

*Code and data availability.* The RHEA code is available in the GitHub repository https://github.com/josebastiase/RHEA and Zenodo repository: https://zenodo.org/record/4767832#.YKKPjyaxVhE. Verification and examples included in this work are found in the examples folder. The Herten analogue data set is available on https://doi.pangaea.de/10.1594/PANGAEA.844167.

*Author contributions.* JMBE developed RHEA, the analytical solutions used for verification, made the figures and tables and wrote the first manuscript draft. GCR closely supervised JMBE. AW provided JMBE with technical support. PB reviewed the manuscript and provided suggestions.

*Competing interests.*   The authors declare that they have no competing interests.

*Acknowledgements.*   This project has received funding from the German Research Council (DFG) grant agreement number 424795466. We would like to thank Mauro Cacace and two anonymous reviewers for their efforts and thoughtful comments that have helped to improve our manuscript.

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
