# Peer review of "RHEA v1.0: Enabling fully coupled simulations with hydro-geomechanical heterogeneity"

_Geoscientific Model Development, 2021_

## Author Comment (AC1)

**Response to Juan Antonio Añel**

We thank the Editor for their supportive comments about our work. Note that we use the abbreviation **EC** for Editor comments and **AR** for our authors' response in the following. Removed text is shown in red, e.g., . New text is shown in blue, e.g., this text has been added.

    **EC:** We have checked your manuscript, and unfortunately, at the moment, it does not comply with our 'Code and Data Policy'. Currently, you archive the code of your model in Github. However, as we state in our policy and Github on its website, it is not a suitable repository for long-term archival.

        **AR:** Dear Juan Antonio Añel, thanks for letting us know about the data storage limitations of Github. We have now migrated RHEA to a long term storage platform. Please find our revisions about the data availability in the manuscript as follows:

        Line 349: Code and data availability.  The RHEA code as well as verification and examples included in this work are available in the Zenodo repository: `https://zenodo.org/record/4767832#.YKKPjyaxVhE`.

---

## Author Comment (AC2)

**Response to Reviewer Comments 1**

We thank the reviewer for their supportive comments about our work. Note that we use the abbreviation **RC1** for reviewer comments 1 and **AR** for our authors' response in the following. Removed text is shown in red, e.g., . New text is shown in blue, e.g., this text has been added.

**RC1:** This is a good description of what sounds like an elegant and useful numerical tool for modelling coupled hydro-geomechanical processes in heterogeneous subsurface environments. I like the model. I think other researchers in the field will find great value in it as well. The modelling results are impressive.

> **AR:** We thank the reviewer for the time and effort in evaluating the manuscript and providing such positive feedback. Please find a detailed response to every comment below. We revised our manuscript as explained by our response.

**RC1:** My point of concern is that the authors, in justifying the raison d'etre of this publication, seem to imply that the list modelling platforms they discuss in the build-up is exhaustive. I know this is not stated explicitly. It is implicit though. In fact, the authors list is, I presume, simply based on their experience. My point is that the list does not have to be exhaustive (in my opinion), but I would advise the authors to state this more clearly.

> **AR:** We agree on that the list of modelling platforms does not have to be exhaustive. Hence, a short disclaimer was added.
>
> Line 38: In the following section, well-known subsurface simulation libraries are briefly reviewed. Since the number of subsurface simulation codes is vast, we only included platforms that are relevant to modelling heterogeneity, for an exhaustive list see White et al. (2018). Current subsurface hydro-geomechanical simulation codes can be classified (...)

**RC1:** I cannot seem to find any statement regarding the computational effort of this code. I would consider this important information.

**AR:** We have added this information to the manuscript as well as the caption of each figure containing simulation results.

Line 171: The four verification scenarios are described in the following subsections. The numerical solution were simulated in a 8 core Intel i7-3770 CPU @ 3.40GHz with 32 GB DDR4 RAM memory and the results were stored on a hard disk drive.

Figure 2. Comparison of analytical and numerical solution of the one dimensional consolidation problem obtained in a sample of 100 m. The lines represent the analytical solution whereas the dots represent the RHEA solution. (a) Homogeneous case. For this simulation, a total of 100 nodes and 99 elements were set. The total time for computing 101 time steps was 1.92 s (b) Heterogeneous case For this simulation, a total of 100 nodes and 99 elements were set. The total time for computing 701 time steps was 13.8 s.

Figure 3. (...) solution which is not feasible to replicate the latter with RHEA. The total simulation time was 312.6 s for 101 time steps and 10.000 elements with 20.502 nodes.

Figure 5. (...) colour range to highlight the small variations in pore pressure. The total simulation time was 0.49 hours for 70 time steps and 44.341 elements with 44.800 nodes.

**RC1:** The examples are in 2D. What is the practical feasibility – both in terms of availability of information/data and in terms of computational effort – of such simulations in 3D?

> **AR:** A three-dimensional setup would not add much complexity to the proposed workflow. The user just needs to add an extra dimensional variable (z-dimension in this case) during the data formatting in the python script. From there, RHEA only needs minor adjustments in the simulation control file. For example, set gravity to the right direction as well as include a third dimension in the mesh. Unfortunately, we cannot say how much the computational efforts would increase, since that would depend not only on the number of elements of the mesh but also the hydro-geomechanical properties of the setup since the mesh will automatically be refined in those nodes where gradients are sharp. For this contribution, only a two-dimensional example was considered for the sake of simplicity as the authors believe this demonstrates RHEA's potential.

If the reader is interested in further insights into RHEA's computational performance, Permann et al. (2020) tested MOOSE under challenging conditions with meshes consisting of more than 78 million nodes. Moreover, the same authors tested the parallel capabilities of MOOSE in thousands of processor cores. Since RHEA is based on MOOSE, it should theoretically be possible to perform any RHEA simulation with at least that number of cores in parallel. We add to the discussion

Line 349: (...) meshing capabilities with implicit time stepping. In this work RHEA capabilities has been studied in two dimensions, but three dimensional simulations are also possible. In that case, a three dimensional mesh according to the data set has to be generated, from this point the data formatting workflow and simulation control are applied as described in section 3.

**RC1:** Line 74: " However, a more robust implementation is Porous Flow, ... " I cannot say whether this statement is true or not, but I suggest that the authors buttress such statements with facts. Why is PorousFlow the more robust implementation and where has this been shown?

> **AR:** The term "robust" was utilized in this context because up to now (17/05/2021) GOLEM neither supports multi-phase flow nor chemical coupling, whereas Porous-Flow does. It is true, however, that GOLEM has been used more extensively than PorousFlow judging by scientific publications, for example (Freymark et al., 2019; Blöcher et al., 2018; Jacquey et al., 2018; Peters et al., 2018). We agree that the sentence may be misleading people who are not familiar with the MOOSE environment.
>
> Line 74 - 75:
>
> Line 74 - 75: Another cutting-edge implementation is PorousFlow, an embedded MOOSE library to simulate multi-phase flow and Thermal-Hydraulic-Mechanical-Chemical (THMC) processes in porous media (Wilkins et al., 2020)

**RC1:** Line 82: I cannot see why the MOOSE naming convention or which kind of bird Rhea is should be relevant to the reader.

**AR:** Agree. The following line has been removed

Line 81 - 82:
Line 81 - 82: The name Rhea depicts a flightless bird that is native to the South American continent.

**References**

G. Blöcher, M. Cacace, A. B. Jacquey, A. Zang, O. Heidbach, H. Hofmann, C. Kluge, and G. Zimmermann. Evaluating micro-seismic events triggered by reservoir operations at the geothermal site of groß schönebeck (germany). *Rock Mechanics and Rock Engineering*, 51(10):3265–3279, 2018.

J. Freymark, J. Bott, M. Cacace, M. Ziegler, and M. Scheck-Wenderoth. Influence of the main border faults on the 3d hydraulic field of the central upper rhine graben. *Geofluids*, 2019, 2019.

A. B. Jacquey, L. Urpi, M. Cacace, G. Blöcher, G. Zimmermann, and M. Scheck-Wenderoth. Far field poroelastic response of geothermal reservoirs to hydraulic stimulation treatment: Theory and application at the groß schönebeck geothermal research facility. *International Journal of Rock Mechanics and Mining Sciences*, 110:316–327, 2018.

C. J. Permann, D. R. Gaston, D. Andrš, R. W. Carlsen, F. Kong, A. D. Lindsay, J. M. Miller, J. W. Peterson, A. E. Slaughter, R. H. Stogner, et al. Moose: Enabling massively parallel multiphysics simulation. *SoftwareX*, 11:100430, 2020.

E. Peters, G. Blöcher, S. Salimzadeh, P. J. Egberts, and M. Cacace. Modelling of multi-lateral well geometries for geothermal applications. *Advances in Geosciences*, 45:209–215, 2018.

M. White, P. Fu, M. McClure, G. Danko, D. Elsworth, E. Sonnenthal, S. Kelkar, and R. Podgorney. A suite of benchmark and challenge problems for enhanced geothermal systems. *Geomechanics and Geophysics for Geo-Energy and Geo-Resources*, 4(1):79–117, 2018.

A. Wilkins, C. P. Green, and J. Ennis-King. Porousflow: a multiphysics simulation code for coupled problems in porous media. *Journal of Open Source Software*, 5(55):2176, 2020.

---

## Author Comment (AC3)

**Response to Reviewer Comments 2**

We thank the reviewer for evaluating our manuscript and for the constructive comments. Note that we use the abbreviation **RC2** for reviewer comments and **AR** for our authors' response in the following. Removed text is shown in red, e.g., . New text is shown in blue, e.g., this text has been added.

**RC2:** I read with great interest the manuscript entitled "RHEA v1.0: Enabling fully coupled simulations with hydro-geomechanical heterogeneity" by Espejo and co-workers.

Despite I acknowledge the efforts from the authors, I am adivising for a major revision before the paper can be consider for its publication in GMD. The main criticism is that the paper lacks the scientific novelty required to make a major contribution to the related community. This is not a limitation of their approach (I think), but rather stems from the authors' choice of the examples discussed in the paper. Indeed, while claiming that their workflow contribute to advance the scientific computing efforts for complex subsurface applications, they limited their discussion to simplistic analytical examples. Terzaghi consolidation is a relatively simple benchmark, which should only be used to validate the physics implementation of their numerical tool. However, this is not the topic of the manuscript, given that the authors relied on existing modules (Porous Flow and Tensor Mechanics). Their realistic example is also a synthetic and simple one, 2 dimensional and with no clear tangible application.

I am saying this, cause the reader is left with the (wrong?) impression that the main contribution of the manuscript is the development of a python script binding MOOSE's objects, which to my opinion does not satisfy the minimum scientific level for a publication (it would rather fit as an internal report). This said, I would warmly advise the authors to re-organize the manuscript in a way to better convene the main message of their work, and, in doing so, to prove their generic sentencing as "... Our work is a valuable tool to assess challenging real world hydro-geomechanical systems that may include different levels of complexity like heterogeneous geology with several time and spatial scales and sharp gradients produced by contrasting subsurface properties. ...", for which I could not find any concept of proof in the text.

**AR:** We agree and therefore we appreciate the opportunity to clarify the novelty of our work.

1. While RHEA utilizes existing modules of the MOOSE framework (namely Porous Flow and Tensor Mechanics), we significantly changed the underlying MOOSE code (directly in C++) to integrate new material objects that, for the first time, unlock modelling of spatially distributed heterogeneity of mechanical in addition to hydraulic properties.

2. To ease pre-processing efforts arising from this improvement, we developed a Python-based, automated workflow which uses standard data format to generate input files that are compatible with the new material objects in MOOSE format. This helps users integrate spatially distributed properties into the modelling workflow.

3. We verified the correctness of our novel application with newly developed, analytical benchmark problems. This includes, for the first time, vertical heterogeneity and illustrated its performance using a sophisticated 2D example with distributed hydraulic and mechanical heterogeneites.

This novel development results in a numerical simulator that we call Real Heterogeneity App (RHEA). To the best of our knowledge, comparable open-source and verified simulators for coupled and fully distributed hydraulic and geomechanical heterogeneities have not been documented in the peer-reviewed scientific literature yet.

Nonetheless, the comment made by the reviewer showed us that the novelty of our work was insufficiently formulated in our manuscript. We therefore take the opportunity to improve our manuscript as follows.

Suggested revision at line 78: "However, it has not yet been extended and verified for the simulation of spatial heterogeneity of mechanical parameters. In other words, despite PorousFlow is able to handle spatially distributed heterogeneity of permeability and porosity, it does not support spatially distributed heterogeneity of mechanical properties such as bulk and Young's moduli."

Suggested revision at line 83: "RHEA is based on MOOSE's modular ecosystem and combines the capabilities of Porous Flow and Tensor Mechanics with material objects that are newly developed in our work and provide the novel ability to allocate spatially distributed properties at element-resolution in the mesh. To achieve that, we significantly changed the underlying MOOSE code of Porous Flow and Tensor Mechanics by integrating new C++ objects. To ease pre-processing efforts arising from this improvement, we developed a Python-based, automated workflow which uses standard data format to generate input files that are compatible with the material objects in MOOSE format. Finally, we verified the correctness of RHEA with a newly developed, analytical benchmark problems allowing vertical heterogeneity and illustrated its performance using a sophisticated 2D example with distributed hydraulic and mechanical heterogeneity."

**RC2:** (...) they limited their discussion to simplistic analytical examples. Terzaghi consolidation is a relatively simple benchmark, which should only be used to validate the physics implementation of their numerical tool.

**AR:** We agree that Terzaghi's problem is a simple benchmark problem, however it provides the opportunity to verify the hydro-geomechanical coupling of numerical solvers, see (Haagenson et al., 2020; Park et al., 2019; Ferronato et al., 2010; Kim et al., 2015; Blanco-Martín et al., 2017; Ma et al., 2017). In our contribution, we decided to verify the hydro-geomechanical coupling of RHEA which is based on combining Porous Flow and Tensor Mechanics as this numerical implementation has not yet been verified in the peer-reviewed scientific literature to the best of our knowledge.

Because we recognise the lack of complexity of Terzaghi's problem, we developed and illustrated two new benchmark examples that increase in complexity. The first uses analytical solutions to describe one-dimensional (vertical) heterogeneity, the second shows a two-dimensional system based on real-world data. We believe that these examples sufficiently verify RHEA and note that developing higher levels of complexity for benchmarking is out of scope for this manuscript (see for example (Green et al., 2021)).

Suggested revision at line 171: "The tests were designed to gradually build up complexity and cover the typical spectrum of consolidation problems. The first test, the classical Terzaghi's problem, is simply used as a basic benchmark of the hydro-mechanical coupling in RHEA. Later sections we illustrate the full possibilities of RHEA on spatially-heterogeneous systems in one and two dimensions.

**RC2:** (...) However, this is not the topic of the manuscript, given that the authors relied on existing modules (Porous Flow and Tensor Mechanics).

> **AR:** While RHEA relies on existing physical modules of the MOOSE framework, we developed new materials that extend the capabilities to those physical modules. Further development of existing capabilities forms one of the fundamental and desired characteristics of the MOOSE framework, see previous MOOSE based numerical solvers that were built in a similar way (Keniley and Curreli, 2019; Slaughter and Johnson, 2017; Jacquey and Cacace, 2017).
>
> Suggested revision at line 7 (Abstract): (...) "element-resolution hydro-geomechanical properties in coupled simulations. To achieve that we developed new materials that extend the capabilities of the current MOOSE physical modules."
>
> Suggested revision at line 81: "The aim of this paper is therefore to develop, verify and illustrate a novel and generic workflow for modelling fully coupled hydro-geomechanical problems allowing the inclusion of hydraulic and geomechanical heterogeneity inherent to realistic geological systems. This was achieved by extending the current capabilities of the MOOSE of two its native physical modules, namely Porous Flow and Tensor Mechanics. We call this workflow RHEA" (...)

**RC2:** (...) Their realistic example is also a synthetic and simple one, 2 dimensional and with no clear tangible application.

> **AR:** The Herten analog is a high resolution hydraulic dataset assembled from a real outcrop, as described in section 4.4.1. We decided to use this well-knwon aquifer analog due to the fact that we could not find any other suitable datasets that integrate full hydro-geomechanical heterogeneity. While we agree that this example is synthetic, the aim was to (1) illustrate how to generate the data files to simulate with RHEA from real data sets, (2) demonstrate increased spatial complexity and sharp gradients. The example could be extended to three dimensions, but it was reduced to two dimensions as this is sufficient to demonstrate RHEA's capabilities and facilitates visualisation of the results.

The aim of this manuscript is to document the development and verification of a simulator for fully distributed heterogeneity, so we believe further modelling of a real problem is beyond the scope of this contribution. We intend to do this in a follow-up study.

Regarding a tangible application, we are currently utilizing RHEA to model land subsidence due to heavy groundwater extraction. We calibrate our models by fitting hydro geomechanical parameters generated by RHEA's workflow to vertical displacement data sets which have fine spatial discretization. RHEA's workflow dramatically simplifies the model fitting since we are able to allocate material properties to each displacement measuring point and the area around it.

Suggested revision at line 272: "The last example aims to study and illustrate the performance of RHEA's with a real data set.  This example illustrates how to generate input files with the developed workflow and demonstrates the potential of RHEA in increased spatial complexity and sharp gradients. While the Herten analog is a 3D data set, the example was reduced to two dimensions to facilitate visualization. Regardless, simulations in three dimensions would not modify the presented workflow. Hence, the 2D consolidation problem was solved with RHEA, integrating the multi-facies realizations and material properties of the Herten analog (Bayer et al., 2015). Although the data set does not contain geomechanical subsurface properties, the hydraulic conductivity varies in six orders of magnitude which represents a sufficient proof of concept for RHEA.

Suggested revision at line 341: "With RHEA a number of future potential applications rise, for instance, land subsidence is a process that can occur due to progressive consolidation due to slow downward percolation over several soil layers due to anthropogenic decrease of the system pressure. The process highly depends on the spatial distribution of the geomechanical properties of the subsurface, RHEA's workflow can potentially simplify such numerical simulations."

Suggested revision at line 345: " However, RHEA could potentially be extended to include also thermal processes and three dimensional simulations."

**RC2:** This said, I would warmly advise the authors to re-organize the manuscript in a way to better convene the main message of their work, and, in doing so, to prove their generic sentencing as "... Our work is a valuable tool to assess challenging real world hydrogeomechanical systems that may include different levels of complexity like heterogeneous geology with several time and spatial scales and sharp gradients produced by contrasting subsurface properties. ...", for which I could not find any concept of proof in the text.

**AR:** The aim of this contribution is to document the development of RHEA and illustrate how realistic geology leading to full and distributed hydro-geomechanical heterogeneity can be modelled. Starting from a relatively simple example (Terzaghi's problem), the level of complexity was increased until a real-world example was presented. We note that sharp gradients can be seen in section 4.2 where hydraulic conductivity varies over four orders of magnitude and are further illustrated in section 4.4 where the hydraulic conductivity varies across six orders of magnitude.

To better reflect this aim, we have revised the statement in line 12 as follows: Our work is a valuable tool to assess challenging real world hydro-geomechanical systems that may include different levels of complexity like heterogeneous geology with  sharp gradients induced by contrasting subsurface properties.

Suggested revision at line 170: "The tests were designed to gradually build up complexity and cover the typical spectrum of consolidation problems. In two of this examples RHEA's performance on simulations with sharp gradients is tested. First, a one dimensional consolidation problem where the hydraulic conductivity varies in four orders of magnitude between layers and a second two dimensional example with realistic heterogeneity in which the hydraulic conductivity of the facies varies in six orders of magnitude."

In terms of organisation, we have attempted to structure our manuscript in a logical way from development to verification with increasing complexity abd therefore we have not changed the organization.

**RC2:** * Abstract - sentence at lines 4-5 - it requires some reworking. Stating that there are no simulations considering heterogeneity in the subsurface is simply not true (an extensive literature research is required here). In addition, it is not clear what they mean by "verification".

**AR:** We agree. For further reference we quote our sentence in line 4-5: "However, implementation and verification of full heterogeneity of subsurface properties using high resolution field data in coupled simulations has not been done before."

We agree that this sentence may not be clear enough. Specifically, the "full" heterogeneity refers to coupled hydraulic AND mechanical heterogeneity, for which there are no open source numerical simulators reported on in the peer-reviewed scientific literature. A close example would be other MOOSE based simulators, where the user can potentially define spatial distributed mechanical parameters by specifying different materials when the mesh is generated (either with MOOSE or other mesh generator software), this workflow works well for simplified systems, for instance layered geology. The latter is illustrated in GOLEM Jacquey and Cacace (2017) which is also a MOOSE-based derivative. However, if realistic geology with complex and naturally shaped geometries is introduced, defining materials when the mesh is generated becomes a significant effort. This is why we introduce our Python workflow in which the user defines location of the properties by specifying Cartesian coordinates as well as the values. The generated files are MOOSE compatible, imported into the simulation and automatically assigned to the appropriate mesh by the dedicated material objects included in RHEA.

Similar workflows have been used in the past, but are limited to coupled hydraulic simulations (only for hydraulic conductivity and porosity, see (Maier et al., 2005; Höyng et al., 2015)). To the best of our knowledge, RHEA provides this capability for coupled hydraulic and geomechanical properties.

We also agree that "verification" may not be an intuitive term to use in this context. Further explanation is added.

Line 32: "However, integrating spatial material properties to numerical tools typically is an arduous task since natural shaped geological formations are made of complex geometries produced by natural processes around them. Advance proprietary numerical tools with a graphical interface such as COMSOL Multiphysics (1998), can format material data to the simulation by importing text files to the software. However in this case, an automatic interpolation between neighbor materials is internally performed, which may lead to unwanted results. To the best of the authors' knowledge there is no open source numerical tool able to integrate full heterogeneity of complex geologic formations for all hydro-geomechanical parameters."

Line 85: "We then compare RHEA's simulation results (verify)  with one and two dimensional analytical solutions, and propose a benchmark semi-analytical solution to investigate RHEA's performance when sharp gradients are present."

**RC2:** * Introduction - sentence at lines 26-27 - also this sentence is not true ("infeasible").

**AR:** We agree and will revise accordingly:

Line 26: (...) "so that fine spatial discretization around fractures is needed in certain numerical models, resulting in  expensive computational demands"

**RC2:** *Introduction - sentence at lines 65-67 - The coupling among physics is not done automatically but via off-diagonal components in the system's Jacobian matrix.

**AR:** We agree and corrected our text accordingly:

Line 65: (...) "each physical process (or its partial differential equation, PDE) is treated separately as an individual MOOSE object and coupling is performed  by the back end routines of MOOSE."

**RC2:** * Paragraph 3 - sentence at line 136 -Material properties are defined at the element level by default in any FEM application.

**AR:** We agree and revised as follows:

Line 136: (...)
(...) "We accomplished this through new MOOSE-based materials functions able to allocate data in each element of the mesh based on a pre-generated input file. "

**RC2:** * Paragraph 3 - sentence at line 155 - Why the size of the mesh needs to match the coverage of the sampling data? Is that a limitation of a naive implementation of the interpolation routine used within the workflow?

**AR:** Since the materials properties have to be assigned at every element, elements with unassigned material properties would raise an error. RHEA automatically linearly interpolates the value of the material property to the unassigned element. Hence, the user has to be careful that the dataset matches the mesh discretization size. Otherwise, interpolated values will be assigned which could lead to significant errors. If further refinement of the mesh is necessary, the mesh adaptivity of MOOSE can be activated, as described in section 4. Hence, we extended our previous statement as follows:
We plan to amend this statement as follows:

Line 156: RHEA may assign unwanted property values,

"since RHEA will automatically linearly interpolate the values provided to the mesh. Thus, if the initial mesh discretization does not match the user-supplied samples, interpolated values are assigned, which could lead to unexpected behavior."

**RC2:** Why rephrasing equation 8 in terms of the hydraulic conductivity? It does not match the formalism used in the previous paragraph ...

**AR:** We agree and will revise accordingly:

Line 106: We express equation 3 in terms of hydraulic conductivity.

$$\boldsymbol{q}_d = \phi(\boldsymbol{v}_f - \boldsymbol{v}_s) = -\frac{\boldsymbol{k}}{\mu_f \rho_f \boldsymbol{g}}(\nabla p_f - \rho_f \boldsymbol{g}) \,, \tag{3}$$

Line 107: "where $\mathbf{v}_f$ and $\mathbf{v}_s$ are the fluid and solid matrix velocities respectively;  $\mathbf{k}$ is the hydraulic conductivity tensor; "

We appreciate the time and effort made by the reviewer and therefore acknowledge him in our acknowledgement.

**References**

L. Blanco-Martín, J. Rutqvist, and J. T. Birkholzer. Extension of tough-flac to the finite strain framework. *Computers & Geosciences*, 108:64–71, 2017.

M. Ferronato, N. Castelletto, and G. Gambolati. A fully coupled 3-d mixed finite element model of biot consolidation. *Journal of Computational Physics*, 229(12):4813–4830, 2010.

C. P. Green, A. Wilkins, J. Ennis-King, and T. LaForce. Geomechanical response due to nonisothermal fluid injection into a reservoir [U+2606]. *Advances in Water Resources*, 153:103942, 2021.

R. Haagenson, H. Rajaram, and J. Allen. A generalized poroelastic model using fenics with insights into the noordbergum effect. *Computers & Geosciences*, 135:104399, 2020.

D. Höyng, H. Prommer, P. Blum, P. Grathwohl, and F. M. D'Affonseca. Evolution of carbon isotope signatures during reactive transport of hydrocarbons in heterogeneous aquifers. *Journal of contaminant hydrology*, 174:10–27, 2015.

A. Jacquey and M. Cacace. Modelling fully-coupled thermo-hydro-mechanical (thm) processes in fractured reservoirs using golem: a massively parallel open-source simulator. In *EGU General Assembly Conference Abstracts*, page 15721, 2017.

S. Keniley and D. Curreli. Crane: A moose-based open source tool for plasma chemistry applications. *arXiv preprint arXiv:1905.10004*, 2019.

J. Kim, E. Sonnenthal, and J. Rutqvist. A sequential implicit algorithm of chemo-thermo-poro-mechanics for fractured geothermal reservoirs. *Computers & geosciences*, 76:59–71, 2015.

T. Ma, J. Rutqvist, C. M. Oldenburg, W. Liu, and J. Chen. Fully coupled two-phase flow and poromechanics modeling of coalbed methane recovery: Impact of geomechanics on production rate. *Journal of Natural Gas Science and Engineering*, 45:474–486, 2017.

U. Maier, A. Becht, B. Kostic, C. Burger, P. Bayer, G. Teutsch, and P. Dietrich. Characterization of quaternary gravel aquifers and their implementation in hydrogeological models. *IAHS PUBLICATION*, 297:159, 2005.

C. Multiphysics. Introduction to comsol multiphysics®. *COMSOL Multiphysics, Burlington, MA, accessed Feb*, 9:2018, 1998.

C.-H. Park, T. Kim, E.-S. Park, Y.-B. Jung, and E.-S. Bang. Development and verification of ogsflac simulator for hydromechanical coupled analysis: Single-phase fluid flow analysis. *Tunnel and Underground Space*, 29(6):468–479, 2019.

A. Slaughter and M. Johnson. Pika: A snow science simulation tool built using the open-source framework moose. In *AGU Fall Meeting Abstracts*, volume 2017, pages C22B–08, 2017.

---

## Author Comment (AC4)

**Response to Reviewer Comments 3**

We thank the reviewer for evaluating our manuscript and for the constructive comments. Note that we use the abbreviation **RC3** for reviewer comments and **AR** for our authors' response in the following. Removed text is shown in red, e.g., . New text is shown in blue, e.g., this text has been added.

**RC3:** The manuscript describes a new contribution to the modelling of hydro-mechanical problems in form of a MOOSE-based simulator. While I think that it deserves publication in GMD, it needs substantial improvements as outlined below. This particularly concerns Section 3 which should be made much more detailed and explicit to be of any value. Moreover, the literature review on existing implementation efforts should be considerably improved. So far, it arbitrarily picks three open-source packages and discredits them with oversimplifying statements.

> **AR:** We apologize for our phrasing regarding the 3 open-source packages. We will expand our literature review of numerical codes, improve formulation about their capabilities and extend Section 3 to include more details.

**RC3:** - l.44ff: I don't agree with this statement. The mentioned references indicate that more sophisticated sequential schemes like iterated fixed-stress perform quite well. It's the "naive" or "straightforward" approaches like drained-split or non-iterated fixed-stress which might perform poorly. Please rephrase.

> **AR:** We agree and will revise accordingly:
>
> Line 44: "Although sequential codes allow flexible and efficient code management in conjunction with reasonable computational costs, they tend to perform poorly in tightly coupled processes, since transient interaction between variables may not be computed accurately (Kim et al., 2011). However, sequential coupling can significantly improve its numerical accuracy when is combined with iterative schemes. In such implementations, feedback between variables is perform by transferring flow variables to the geomechanics then inserting calculated stress and strain back into the flow problem for the next iteration (Beck et al., 2020). Stability of the iterative methods is discussed by Kim et al. (2011); Mikelić and Wheeler (2013).

**RC3:** - l.55ff: There's a more recent "official" paper on PorePy which is better to cite than the 2017 one: https://link.springer.com/article/10.1007/s10596-020-10002-5. I think that the statement "these codes are in an immature stage" doesn't properly reflect the apparently

rather big efforts behind, at least, PorePy. I'm also not sure about the argument with the "point and line sinks", maybe they have already been integrated or maybe it's very easy to integrate them as a user of the framework. To me, the intention of these codes is just very different from company efforts like COMSOL or coupling frameworks such as MOOSE and probably not what the authors need or aim for.

> **AR:** We agree and will revise this accordingly. In our opinion PorePy is a novel open source software for fast prototyping of modelling tools that can later on be applied to more general frameworks such as MOOSE or FEniCS, Dune, etc.
>
> Suggested revision at line 55: "The second is called Porepy and was specifically developed to simulate HM processes in rock fractures Keilegavlen et al. (2021). Despite the fact that python-based coding offers the advantage of high-level programming within a relatively friendly user interface,  these codes are design to facilitate rapid development of features that cannot be properly represented by standard simulation tools rather than general multiphysics problems.."

**RC3:** - l.60f: I also don't like the negative statements here or at least the general way in which they are formulated. I'm sure that the developers of OpenGeoSys would disagree. It's open source, anyone can adapt the "fundamental governing equations" to her needs. In principle. Please relate things with something like "from our experience, it's rather difficult to..."

> **AR:** Thank you for identifying our negativity! We agree and will revise as follow:
>
> Suggested revision at line 60: "The code is well documented and features several examples in different subsurface areas,  further new developers are constantly contributing with new features to the source code (Graupner et al., 2011; Kosakowski and Watanabe, 2014; Li et al., 2014). From our experience, however, users without some computer science background, experience programming in C++ or python and experience using a debugger, may require a significant amount of time to take full advantage of the features that OGS offers. Furthermore, the authors are unaware of a peer-reviewed verification which includes full geomechanical heterogeneity."

**RC3:** - The list of other mentioned packages is very short and subjective. There are many other efforts for hydro-mechanical modelling based on other frameworks, such as

\* https://doi.org/10.1016/j.cam.2016.10.022 or https://doi.org/10.1142/S1793962321500033 based on deal.II

\* https://doi.org/10.1007/s10596-020-09987-w based on Dumux

\* https://doi.org/10.1029/2019JB017298 based on MRST

\* PFLOTRAN can do geomechanics

All of these frameworks can deal with heterogeneous material parameters.

> **AR:** Thank you for providing these references. While we agree that the the list of numerical codes could be made more exhaustive, none of the suggested codes can integrate spatially distributed heterogeneity of both hydraulic and geomechanical properties like RHEA does. To the best of our knowledge, all other codes can merely integrate spatial geomechanical heterogeneity in the form of layers or as pre-defining material blocks in the mesh which makes the workflow tedious. RHEA significantly simplifies this task by allowing the user to pre-define spatially distributed properties in the Python environment.
>
> Suggested revision at line 44: Another massively parallel subsurface flow code PFLORTRAN, an open-source, multiscale and multiphysics code for subsurface and surface processes (Hammond et al., 2014). The code solves for non-isothermal multiphase flow, reactive transport and geomechanics in porous medium. PFLOTRAN has been applied to model hydro geomechanical systems (Lichtner and Karra, 2014).
>
> Suggested revision at line 59: MRST is an open-source code developed for fast prototyping of new tool in reservoir modelling (Lie, 2019) coded on MATLAB. Although MRST is not a simulator itself, supports multyphase flow with THM physics. MRST has been used in hydro geomechanical in fracture rock in the past (Zhao and Jha, 2019) as well as several other subsurface applications (Garipov et al., 2018; Ahmed et al., 2017; Edwards et al., 2017)
>
> Suggested revision at line 62: DuMux is a fully coupled numerical simulator for multi phase flow and transport in porous medium, free and open source (Flemisch et al., 2011). It is based on the Distributed Unified Numeric Environments (DUNE) which is a C++ based ecosystem to solve FEM based on PETSc solvers. DuMux is well known for its strong focus on multi-phase flow and transport in porous media, its recent realise adds extra features which facilitates physics coupling such as Navier-stokes models (Koch et al., 2021).
>
> Suggested revision at line 67: Other novel codes include (Dang and Do, 2021; Tran and Jha, 2020; Reichenberger, 2003; Martin et al., 2005; Frih et al., 2012)

**RC3:** - l.70f: "for example an experienced user can easily modify the source code to add desired features such as..." I'm convinced that this also holds for the open-source codes which are mentioned above and which have been put in a negative light.

> **AR:** We agree and will revise accordingly.
>
> Suggested revision at line 67: We have found that mastering the basic concepts of the MOOSE workflow requires a steep learning curve. However, it requires minimum C++ coding skills which facilitates the learning experience from users that not necessarily have a computer science background. Once the basics are mastered the benefits are significant, for example an experienced user can easily modify the source code to add desired features such as multi-scale physics, non-linear material properties, complex boundary conditions or even basic post-processing tools with only a few lines of code.

**RC3:** - Section 3 does a very poor job in describing the implementation of the model, as it doesn't connect enough to Section 2. There's only a few lines 135ff which make an explicit connection, the rest is generic blabla. I consider this the most important section of the manuscript. Please be much more precise here. How are the two modules integrated? How are they coupled? What finite elements are used? What time integration scheme? What about local mass conservation?...

> **AR:** We have added the following material to the end of Section 2, in an attempt to succinctly describe these features.
>
> As derivative of the MOOSE framework, RHEA has access to a wide array of options for tuning a simulation. Solver options such as numerical schemes, adaptive timestepping, and PETSc options are available. By default, RHEA uses a first order fully-implicit time integration (backward Euler) for unconditional stability and solves the above equations simultaneously (full coupling) (Kavetski et al., 2002; Manzini and Ferraris, 2004; Gaston et al., 2009). RHEA also allow operator splitting to implement loose coupling (solving the fluid flow while keeping the mechanics fixed, then solving the mechanics while keeping the fluid-flow fixed) and the latter can even occur on separate meshes with different time-stepping schemes, but this feature is not explored in the current article (Martineau et al., 2020)

Explicit time integration (with full or loose coupling) and other schemes such as Runge-Kutta are available in MOOSE and RHEA, but stability limits the time-step size, so these are rarely used in the type of subsurface problems handled by RHEA. By default, MOOSE and RHEA use linear Lagrange finite elements (tetrahedra, hexahedra and prisms for 3D problems, triangles and quads for 2D problems), but higher-order elements may be easily chosen if desired (Hu, 2017) RHEA does not implement any numerical stabilization for the fluid equation to eliminate overshoots and undershoots, however, fluid volume is conserved at the element level (Cacace and Jacquey, 2017). Although not explored in this article, RHEA's fluid flow may be extended to multi-phase, multi-component flow with high-precision equations of state, as well as finite-strain elasto-plasticity (Wilkins et al., 2020).

**RC3:** - l.349: I saw that you are required to link to Zenodo. That's ok, but please keep also the link to GitHub, that's where your code is developed further, hopefully.

**AR:** Done.

**RC3:** - l.67: What are "gold-standard numerical solvers"?

**AR:** We will explain this in our revised version.

Suggested revision at line 67: "It offers  clean and effective numerical PDEs solvers as well as mesh capabilities with a uniform approach for each class of problem. This design enables easy comparison and use of different algorithms (for example, to experiment with different Krylov subspace methods, preconditioners, or truncated Newton methods) which are under constant development.

**RC3:**
l.32, 92, 102, ...: a single "porous mediUM", many porous media
- l.161: "designED"
- l.189: "undraiNed"
- (19): "L" instead of "H"
- l.336: "play" without "s"

**AR:** Corrected.

**References**

E. Ahmed, J. Jaffré, and J. E. Roberts. A reduced fracture model for two-phase flow with different rock types. *Mathematics and Computers in Simulation*, 137:49–70, 2017.

M. Beck, A. P. Rinaldi, B. Flemisch, and H. Class. Accuracy of fully coupled and sequential approaches for modeling hydro-and geomechanical processes. *Computational Geosciences*, 24(4):1707–1723, 2020.

M. Cacace and A. B. Jacquey. Flexible parallel implicit modelling of coupled thermal–hydraulic–mechanical processes in fractured rocks. *Solid Earth*, 8(5):921–941, 2017.

H.-L. Dang and D.-P. Do. Finite element implementation of coupled hydro-mechanical modeling of transversely isotropic porous media in deal. ii. *International Journal of Modeling, Simulation, and Scientific Computing*, 12(01):2150003, 2021.

R. W. Edwards, F. Doster, M. A. Celia, and K. W. Bandilla. Numerical modeling of gas and water flow in shale gas formations with a focus on the fate of hydraulic fracturing fluid. *Environmental science & technology*, 51(23):13779–13787, 2017.

B. Flemisch, M. Darcis, K. Erbertseder, B. Faigle, A. Lauser, K. Mosthaf, S. Müthing, P. Nuske, A. Tatomir, M. Wolff, et al. Dumux: Dune for multi-{phase, component, scale, physics,...} flow and transport in porous media. *Advances in Water Resources*, 34(9): 1102–1112, 2011.

N. Frih, V. Martin, J. E. Roberts, and A. Saâda. Modeling fractures as interfaces with nonmatching grids. *Computational Geosciences*, 16(4):1043–1060, 2012.

T. T. Garipov, P. Tomin, R. Rin, D. V. Voskov, and H. A. Tchelepi. Unified thermo-compositional-mechanical framework for reservoir simulation. *Computational Geosciences*, 22(4):1039–1057, 2018.

D. Gaston, C. Newman, G. Hansen, and D. Lebrun-Grandie. Moose: A parallel computational framework for coupled systems of nonlinear equations. *Nuclear Engineering and Design*, 239(10):1768–1778, 2009.

B. J. Graupner, D. Li, and S. Bauer. The coupled simulator eclipse–opengeosys for the simulation of co2 storage in saline formations. *Energy Procedia*, 4:3794–3800, 2011.

G. E. Hammond, P. C. Lichtner, and R. Mills. Evaluating the performance of parallel subsurface simulators: An illustrative example with pflotran. *Water resources research*, 50 (1):208–228, 2014.

R. Hu. A fully-implicit high-order system thermal-hydraulics model for advanced non-lwr safety analyses. *Annals of Nuclear Energy*, 101:174–181, 2017.

D. Kavetski, P. Binning, and S. W. Sloan. Adaptive backward euler time stepping with truncation error control for numerical modelling of unsaturated fluid flow. *International Journal for Numerical Methods in Engineering*, 53(6):1301–1322, 2002.

E. Keilegavlen, R. Berge, A. Fumagalli, M. Starnoni, I. Stefansson, J. Varela, and I. Berre. Porepy: An open-source software for simulation of multiphysics processes in fractured porous media. *Computational Geosciences*, 25(1):243–265, 2021.

J. Kim, H. A. Tchelepi, and R. Juanes. Stability and convergence of sequential methods for coupled flow and geomechanics: Fixed-stress and fixed-strain splits. *Computer Methods in Applied Mechanics and Engineering*, 200(13-16):1591–1606, 2011.

T. Koch, D. Gläser, K. Weishaupt, S. Ackermann, M. Beck, B. Becker, S. Burbulla, H. Class, E. Coltman, S. Emmert, et al. Dumux 3–an open-source simulator for solving flow and transport problems in porous media with a focus on model coupling. *Computers & Mathematics with Applications*, 81:423–443, 2021.

G. Kosakowski and N. Watanabe. Opengeosys-gem: a numerical tool for calculating geo-chemical and porosity changes in saturated and partially saturated media. *Physics and Chemistry of the Earth, Parts A/B/C*, 70:138–149, 2014.

D. Li, S. Bauer, K. Benisch, B. Graupner, and C. Beyer. Opengeosys-chemapp: a coupled simulator for reactive transport in multiphase systems and application to co 2 storage formation in northern germany. *Acta Geotechnica*, 9(1):67–79, 2014.

P. C. Lichtner and S. Karra. Modeling multiscale-multiphase-multicomponent reactive flows in porous media: Application to co2 sequestration and enhanced geothermal energy using pflotran. In *Computational Models for CO2 Geo-sequestration & Compressed Air Energy Storage*, pages 121–176. CRC Press, 2014.

K.-A. Lie. *An introduction to reservoir simulation using MATLAB/GNU Octave: User guide for the MATLAB Reservoir Simulation Toolbox (MRST)*. Cambridge University Press, 2019.

G. Manzini and S. Ferraris. Mass-conservative finite volume methods on 2-d unstructured grids for the richards' equation. *Advances in Water Resources*, 27(12):1199–1215, 2004.

V. Martin, J. Jaffré, and J. E. Roberts. Modeling fractures and barriers as interfaces for flow in porous media. *SIAM Journal on Scientific Computing*, 26(5):1667–1691, 2005.

R. Martineau, D. Andrs, R. Carlsen, D. Gaston, J. Hansel, F. Kong, A. Lindsay, C. Permann, A. Slaughter, E. Merzari, et al. Multiphysics for nuclear energy applications using a cohesive computational framework. *Nuclear Engineering and Design*, 367:110751, 2020.

A. Mikelić and M. F. Wheeler. Convergence of iterative coupling for coupled flow and geomechanics. *Computational Geosciences*, 17(3):455–461, 2013.

V. Reichenberger. *Numerical simulation of multiphase flow in fractured porous media*. PhD thesis, 2003.

M. Tran and B. Jha. Coupling between transport and geomechanics affects spreading and mixing during viscous fingering in deformable aquifers. *Advances in Water Resources*, 136:103485, 2020.

A. Wilkins, C. P. Green, and J. Ennis-King. Porousflow: a multiphysics simulation code for coupled problems in porous media. *Journal of Open Source Software*, 5(55):2176, 2020.

X. Zhao and B. Jha. Role of well operations and multiphase geomechanics in controlling fault stability during co2 storage and enhanced oil recovery. *Journal of Geophysical Research: Solid Earth*, 124(7):6359–6375, 2019.

---

## Author Response (AR2)

**Response to Sergey Gromov**

We thank the editor Sergey Gromov for his thoughtful comments and verdict in response to our revised manuscript. In the following, we address the remaining concerns. Please note that we use the abbreviation **EC** for editor comments and **AR** for authors' response in what follows. Removed text is shown in red, e.g., . New text is shown in blue, e.g., this text has been added.

**EC:** Thank you very much for preparing the revised manuscript, which in my opinion got certainly improved after you have sufficiently addressed referees' comments. I find the issue of selecting the complexity of the examples presented/statements regarding RHEA capabilities (one of the major concerns raised by one of the Reviewers) is important however likely out of 'Development and technical' scope of this paper – it should certainly be addressed/demonstrated in subsequent studies. I suggest clarifying a few minor and technical remarks (listed below, line nos. refer to the Author's tracked changes manuscript, gmd-2021-45-ATC1.pdf), after which we proceed to the publication in GMD.

> **AR:** We thank the editor for the time and effort in evaluating the manuscript and for providing constructive feedback. Please find a detailed response to every comment below.

**EC:** L124-125: You are very welcome to keep the statement regarding the name Rhea in the model manual/website, however there is no context for this statement in this manuscript (furthermore, the abbreviation RHEA is introduced and is sound). Please remove.

> **AR:** We agree and have deleted this statement in the manuscript.
>
> Line 124-125:

**EC:** L134 "results (verify)" is somewhat ambiguous; is there any intention in using parentheses here? For the Introduction section, a mere statement "evaluate RHEA 1D and 2D simulation results" (even omitting details regarding analytical solutions) will suffice.

> **AR:** We agree and have revised this.
>
> Line 134: We then compare RHEA's simulation results  with one and two dimensional analytical solutions, (...)

**EC:** L345,388,392,393: Please use consistent numbering according to GMD requirements (there are stops and commas used in reported nodes/elements nos.).

**AR:** We agree and have revised this accordingly.

**EC:** L352-353: I suggest using "facilitate presentation" or similar here – for the sake of argument, one may facilitate visualisation of a simulated 3D field by plotting a mere 2D section of it.

**AR:** We agree and have changed this.

Line 352-353: While the Herten analog is a 3D data set, the example was reduced to two dimensions to facilitate  presentation.

**EC:** L438: Does not "and three dimensional simulations" reiterate the statement of the last sentence in previous paragraph?

**AR:** Agree and have changed this accordingly.

Line 438: Our current work focuses on hydro-geomechanical coupling of heterogeneous systems. However, RHEA could potentially be extended to include also thermal processes. .

**EC:** L441-442: Please rephrase the statement indicating which repository was used for results presented in the manuscript. E.g., "The code and examples presented in this study are available at Zenodo repository(..). The continuous development of RHEA code is maintained at the GitHub repository (...)."

**AR:** We agree and have clarified this in the revised manuscript.

Line 441-442: .

The code and examples presented in this study are available at Zenodo repository: `https://zenodo.org/record/4767832#.YKKPjyaxVhE`. The continuous development of RHEA code is maintained at the GitHub repository `https://github.com/josebastiase/RHEA`. The Herten analogue data set is available on `https://doi.pangaea.de/10.1594/PANGAEA.844167`.

**EC:** Non-public comments to the Author: L451 The Author's tracked changes manuscript does not have added/removed references marked up – I did not check References section thoroughly.

**AR:** Thanks. We have added the references to the tracked changes version.

---

## Author Response (AR3)

**Response to Sergey Gromov**

Dear Sergey,

Thanks for clarifying our mistake in the notation. We have submitted a new version of the manuscript with the requested corrections.

Kind regards,

the authors.